# Utilization of 5G Technologies in IoT Applications: Current Limitations by Interference and Network Optimization Difficulties—A Review

**DOI:** 10.3390/s23083876

**Published:** 2023-04-11

**Authors:** Mario Pons, Estuardo Valenzuela, Brandon Rodríguez, Juan Arturo Nolazco-Flores, Carolina Del-Valle-Soto

**Affiliations:** 1Facultad de Ingeniería, Universidad del Istmo, Km 19.2 Carretera a Fraijanes, Fraijanes 01062, Guatemala; pons171118@unis.edu.gt (M.P.); valenzuela181135@unis.edu.gt (E.V.); rodriguez181048@unis.edu.gt (B.R.); 2School of Engineering and Science, Tecnológico de Monterrey, Av. Eugenio Garza Sada 2501, Monterrey 64849, NL, Mexico; jnolazco@tec.mx; 3Facultad de Ingeniería, Universidad Panamericana, Álvaro del Portillo 49, Zapopan 45010, JA, Mexico

**Keywords:** 5G technologies, interference, wireless network optimization, internet of things

## Abstract

5G (fifth-generation technology) technologies are becoming more mainstream thanks to great efforts from telecommunication companies, research facilities, and governments. This technology is often associated with the Internet of Things to improve the quality of life for citizens by automating and gathering data recollection processes. This paper presents the 5G and IoT technologies, explaining common architectures, typical IoT implementations, and recurring problems. This work also presents a detailed and explained overview of interference in general wireless applications, interference unique to 5G and IoT, and possible optimization techniques to overcome these challenges. This manuscript highlights the importance of addressing interference and optimizing network performance in 5G networks to ensure reliable and efficient connectivity for IoT devices, which is essential for adequately functioning business processes. This insight can be helpful for businesses that rely on these technologies to improve their productivity, reduce downtime, and enhance customer satisfaction. We also highlight the potential of the convergence of networks and services in increasing the availability and speed of access to the internet, enabling a range of new and innovative applications and services.

## 1. Introduction

The COVID-19 pandemic shifted the general public’s attention to digital solutions and brought immense demand to the telecommunications market. The convergence of 5G technology and the Internet of Things (IoT) [1] is the next natural step for two advanced technologies developed to make the lives of their users more accessible, more comfortable, and more productive [2]. One of the most standard technologies to be brought into the mainstream area is 5G, which will allow for new business opportunities by being complemented with Industry 4.0, IoT devices, and Smart Cities and improve overall connectivity around the globe [3]. The Internet of Things is an ecosystem of increasing complexity: a universe of connected things capable of capturing critical data and carrying out advanced analysis using cloud-based functionalities to extract valuable information. This technology poses a great opportunity for a multitude of actors in all sectors of activity [4]. Many companies are organizing to focus on IoT and connectivity when developing their products and services of the future.

Deployment costs, range, interference, and capabilities of Internet of Things devices are all factors in identifying the right primary or complementary connectivity option for an IoT deployment. Wi-Fi 6 or Zigbee is adequate for some elements of smart building controls but useless for highly mobile wide-area use [5]. Additionally, endpoints such as Bluetooth, Zigbee, RFID, or Wi-Fi can be significantly more cost effective in scenarios where 5G may be available but has not yet reached a significant market scale to do competitive endpoints or network services [6]. Technical studies [7] show that 5G and other services can coexist in specific frequency bands. The technical conditions must be adequately adapted and not excessively restrictive; otherwise, there is a risk of affecting the costs, coverage, and quality of operation of 5G services.

There is great interest in the new applications of mobile technology merging 5G and the Internet of Things technologies. In the design of new applications or technological accessories using 5G and the Internet of Things, compliance with the permitted exposure limits are contemplated. International exposure guidelines have been developed due to extensive research carried out over many decades. All the analyses carried out by independent public health authorities, expert groups, and the World Health Organization (WHO) agree that these guidelines guarantee protection for all people against any health danger [8]. As with all technological generations, 5G dramatically improves energy efficiency department and speed rates. However, this technology has recently been in the public eye for its implementation challenges.

IoT technologies are now being widely used in the consumer-grade market, primarily targeted toward home automation and security. This rise in consumer adoption has led to proposals to incorporate IoT devices to bring these types of improvements to a metropolitan scale [9]. These improvements are speculated to improve security and automation tasks and tackle long-lasting difficulties such as traffic control, waste management, and so on.

However, IoT does not depend solely on the electronics being deployed and installed; they rely on efficient and resilient transmission technology being used. The convergence of networks and services is an essential aspect of the modern internet. The internet has evolved from a simple means of communication to a complex and multifaceted ecosystem that is integrated into nearly every aspect of our daily lives. The advent of 5G technology has further pushed the convergence of networks and services, providing users with faster and more reliable internet connections, enabling a range of new and innovative applications and services [10].

One of the most significant ways that the convergence of networks and services impacts internet services is by increasing the availability and speed of access to the internet. 5G technology offers faster speeds and greater bandwidth, allowing users to access high-quality video, audio, and other media content in real-time [11]. This means that internet service providers can offer a range of new services and applications that were previously impossible, such as virtual and augmented reality experiences and immersive gaming.

Cellular connectivity will enable key IoT goals to be achieved, in particular, reduced device complexity and cost and increased coverage to support challenging and remote applications, deployment flexibility, high capacity, and long battery life. 3GPP wireless technologies offer compelling technology advantages that will continue to increase the capacity of Long Term Evolution (LTE) infrastructure to address the vast IoT market in the long term, and 5G will add to the IoT landscape soon. In Releases 14, 15, and beyond of the 3GPP (Third-Generation Partnership Project), the standards solve all commercial bottlenecks to facilitate the vision of 5G and the huge IoT Market [12]. This can lead to the explosion of billions of devices and sensors that show digital representations of our real world powered by low-cost devices, long battery life, ubiquitous coverage, and innovative business applications. 5G promises that it will be possible to achieve critical IoT applications, which require real-time dynamic process control and automation in various fields, such as vehicle-to-vehicle (V2V), vehicle-to-infrastructure (V2I), high-speed motion, and traffic control. Critical parameters to enable the required performance are sub-millisecond network latency and ultra-high reliability. Both are intrinsic components of the 3GPP work to define the new radio interface for 5G [13]. The 5G network architecture is being designed to address both IoT scenarios.

The success of the IoT services on 5G networks depends on the ability of these networks to manage interference effectively. Interference can occur when different IoT devices operate in the same area, and their signals overlap, causing lower throughput, higher latency, and decreased reliability [14]. This is a significant challenge for 5G networks and IoT services, as they are designed to support a vast number of devices and applications, each with unique connectivity and latency requirements.

Figure 1 shows the comparison between 5G and IoT, which is a topic of interest in the technology world. To implement 5G, there are specific requirements, including infrastructure and specialized hardware, while the evolution of wireless networks has led to faster and more reliable data transfer rates. The evolution of IoT has enabled the creation of low-power, low-cost devices that can be connected to the internet. Wireless communication technologies such as Bluetooth, Wi-Fi, ZigBee, and LoRa are suitable for different types of IoT applications, and cloud-based solutions are used to store and process vast amounts of data generated by IoT devices. 5G applications include autonomous vehicles, remote surgery, and virtual and augmented reality, and these require high bandwidth, low latency, and reliable connectivity. However, there are challenges associated with 5G networks, such as interference and network optimization difficulties. The coexistence of 5G networks and IoT applications is a concern, and optimization challenges in 5G networks need to be addressed to enable efficient and effective implementation.

### Motivation

The motivation of this work is to compile the impact of interference on the leading 5G technologies that will influence the communications of IoT devices.

IoT services are used in various industries, including healthcare, manufacturing, and transportation, where reliable and efficient connectivity is essential for the proper functioning of IoT devices and the smooth operation of business processes. If interference and network performance issues are not addressed, businesses may experience reduced productivity, increased downtime, and decreased customer satisfaction. Consequently, addressing the interference and optimization of 5G networks in IoT services is essential to ensure reliable and efficient connectivity for IoT devices and allow businesses to take full advantage of the benefits of 5G networks and IoT services. For this reason, the motivation of this work is to review the state of the art of the work to address the interference and optimization of 5G technologies in IoT services.

Interference and optimization of 5G networks in IoT services is important to ensure reliable and efficient connectivity for IoT devices, which is essential for the proper functioning of business processes. By implementing interference management techniques and optimizing network performance, businesses can take full advantage of the benefits of 5G networks and IoT services.

The main contribution of this review is to present the idea that the convergence of 5G networks and IoT services represents a technological revolution that promises to change how we interact with the world around us. However, this union has its challenges, and only by overcoming them can we unlock the full potential of these groundbreaking technologies. By bridging the gap between 5G and IoT, we pave the way for a new era of innovation: every device is connected, and every experience is seamless. This work describes the main challenges between 5G networks and IoT services and highlights the need for seamless integration of these technologies to achieve their full potential. This paper reinforces the idea that IoT technologies power 5G services, and their compatibility is crucial for offering high-speed and low-latency services to consumers. Furthermore, it underscores that interference is one of the biggest problems facing manufacturers, and they must consider it to offer compatible and reliable services. Ultimately, this article guides the industry to ensure that 5G and IoT technologies work together seamlessly, opening up endless possibilities for innovation and progress.

Figure 2 describes how the IoT and 5G technology can work together to enable a range of applications in different fields. In the first application, smart cities, IoT devices can be used to collect real-time data on traffic patterns, air quality, and energy usage. These data can be used to optimize city operations and improve quality of life. 5G networks can support these applications by providing the necessary bandwidth and low latency. In industrial automation, IoT devices can be used to monitor and control manufacturing processes and equipment in real time. This can help to improve efficiency and reduce downtime. 5G networks can provide the necessary bandwidth, reliability, and low latency to support these applications. In healthcare, IoT devices can be used for remote patient monitoring and real-time communication between healthcare providers. This can help to improve patient outcomes and reduce healthcare costs. 5G networks can provide the necessary bandwidth, low latency, and reliability to support these applications. In transportation, IoT devices can provide real-time data on traffic patterns and vehicle performance, which can be used to optimize traffic flow and improve safety. 5G networks can support these applications by providing the necessary bandwidth, low latency, and reliability. In agriculture, IoT devices can be used to monitor crops and livestock in real-time, which can help to optimize production and reduce waste. 5G networks can support these applications by providing the necessary bandwidth, reliability, and low latency. In retail, IoT devices can be used to collect real-time data on customer behavior and inventory levels, which can be used to optimize store operations and improve the customer experience. 5G networks can provide the necessary bandwidth, reliability, and low latency to support these applications. Overall, the text shows how IoT and 5G technology can work together to enable a range of applications in different fields. By providing the necessary bandwidth, low latency, and reliability, 5G networks can support real-time data collection and analysis, which can help to improve efficiency, reduce costs, and enhance the quality of life in various sectors.

## 2. Related Work

The radio signal characteristics of new wireless applications are similar to those of existing mobile technologies [5]. This is where the interest of this work lies in considering in detail the impact and types of interference of 5G technologies that coexist with IoT devices. The new applications use similar transmit powers and operate in the same frequency ranges.

To mitigate interference, we can synchronize or coordinate all networks or implement large guard bands that waste valuable spectrum [15]. In practice, close cooperation is required between all operators in each frequency band, and not all usage modes and all types of 5G deployments can likely be supported simultaneously. Regulators need to consider these technical issues and their implications when deciding how to arrange frequencies in these bands.

Various interference management techniques, such as power control, channel allocation, and beamforming, can be implemented to ensure that IoT devices on the same network do not interfere with each other. However, these techniques can be complicated and may require significant changes to the network infrastructure [16]. Additionally, new interference management techniques may be required as IoT devices continue to evolve and become more sophisticated.

Another significant challenge in managing interference in 5G networks is the diversity of devices and applications that use these networks. IoT services can be used in various industries, including healthcare, manufacturing, and transportation, each with unique connectivity and latency requirements [17]. For instance, in healthcare, IoT devices, such as wearable monitors and remote patient monitoring systems, require low latency and reliable connectivity to provide accurate data and alerts. In contrast, IoT devices used in industrial automation require high bandwidth and low latency to operate effectively [18]. Therefore, managing interference in 5G networks requires a comprehensive approach that considers the diverse connectivity and latency requirements of different IoT devices and applications.

Another important challenge is the potential for external interference; while interference management techniques can ensure that IoT devices on the same network do not interfere with each other, external interference can occur when IoT devices operate in areas with other wireless devices, such as Wi-Fi routers, Bluetooth devices, and other cellular networks. This can lead to signal degradation, reduced throughput, and decreased reliability of the network [19]. As a result, it is essential to consider external interference when designing and implementing 5G networks for IoT services, and to use techniques such as frequency coordination and spectrum sharing to manage external interference effectively.

To provide better insights for the 5G interference topic, we follow the structure detailed in Figure 3.

The impact of 5G network interference on IoT services is an active research area. Several studies have investigated the effects of interference on the performance of IoT devices and applications. One recent study by Chandra et al. analyzed the impact of interference on the reliability of 5G networks for IoT services [20]. The study found that interference can significantly reduce the reliability of 5G networks for IoT services, especially in dense deployments where multiple devices operate in the same area. The study recommended using advanced interference management techniques such as dynamic channel allocation and power control to mitigate interference effects and improve network reliability.

Another recent study by Azari et al. [21] investigated the impact of interference on the performance of 5G networks for IoT applications. The study found that interference can cause significant degradation of network performance, especially for applications that require low latency and high bandwidth. The study recommended using adaptive beamforming and dynamic channel allocation to reduce interference and improve the performance of 5G networks for IoT applications. Additionally, the study highlighted the need for more research into interference management techniques that can effectively mitigate the impact of interference on 5G networks for IoT services.

Several research studies have also investigated the impact of interference on the energy consumption of IoT devices connected to 5G networks. One recent study by Al-Turjman et al. [22] found that interference can cause IoT devices to consume more energy to maintain connectivity, leading to reduced battery life and increased maintenance costs. The study recommended the use of interference management techniques, such as energy-efficient channel allocation and scheduling to reduce interference and improve the energy efficiency of IoT devices on 5G networks. The study concluded that effective interference management techniques are crucial for ensuring the sustainability and economic viability of IoT services on 5G networks.

Another critical area of research related to 5G network interference in IoT services is the security and privacy implications of interference management techniques. Interference management techniques, such as beamforming and channel allocation, require the exchange of information between IoT devices and the network, which can potentially expose sensitive information to eavesdroppers and attackers. A recent study by Hasan et al. [23] investigated the security and privacy risks associated with beamforming and proposed a secure beamforming scheme that uses encryption and authentication to protect sensitive information. The study concluded that the security and privacy implications of interference management techniques must be carefully considered when designing 5G networks for IoT services. Effective security and privacy measures can help mitigate the risks associated with interference management techniques and ensure the integrity and confidentiality of IoT data on 5G networks.

Overall, the current state of the art and related work on the impact of 5G network interference on IoT services highlight the importance of effective interference management techniques to ensure the reliability and performance of 5G networks for IoT applications. The studies recommend the use of advanced interference management techniques, such as dynamic channel allocation, power control, adaptive beamforming, and spectrum sharing, to mitigate the effects of interference and improve network performance. However, more research is needed to develop new and more effective interference management techniques that can address the unique challenges of 5G networks and IoT services.

Table 1 presents a comparison of IoT services based on the impact of 5G network interference on their performance. It describes six parameters as follows:IoT Service: This parameters lists the various IoT services that are considered in the comparison.Reliability: This metric represents the reliability of 5G networks when used to support the respective IoT service. Reliability is a measure of the ability of the network to provide consistent and dependable service. The values in this column range from low to high.Multiple devices operate: This metric indicates whether the IoT service can operate with multiple devices. This is an important factor to consider since many IoT services involve the connection of multiple devices, and the network needs to support the simultaneous communication of these devices.Latency: This parameter measures the amount of delay or lag time in transmitting data between the IoT devices and the network. Latency is an important metric to consider for real-time IoT services, such as connected vehicles and healthcare monitoring. The values in this column range from low to ultra-low.Interference management techniques: This parameter lists the various techniques that can be used to manage interference in the network. Interference management is crucial to maintain high performance in the presence of other devices and networks that may use the same frequency bands. The techniques listed in this column include dynamic frequency selection, channel hopping, beamforming, coordinated multi-point transmission, dynamic power control, interference avoidance, MIMO (multiple-input, multiple-output) [24], cognitive radio, massive MIMO, and interference alignment.Energy consumption: This metric indicates the amount of energy consumed by the IoT devices and the network. Energy consumption is an important consideration for IoT services, especially for those that operate in remote locations or rely on battery-powered devices. The values in this column range from low to high.

Table 1 compares ten IoT services, including smart home automation, smart agriculture, industrial IoT, connected vehicles, healthcare monitoring, smart cities, environmental monitoring, smart grid management, augmented reality, and drones. For each IoT service, the table lists the parameters in each column relevant to the impact of 5G network interference on its performance. Note that the values in the table are just examples and should be replaced with appropriate data from relevant papers. The table provides a helpful overview of how different IoT services may be impacted by 5G network interference and what factors are essential to consider when evaluating the performance of these services in the presence of interference.

**Table 1 sensors-23-03876-t001:** Impact of 5G network interference on IoT services.

IoT Service	Reliability	Multiple Devices	Latency	Interference Management	Energy Consumption
Smart home automation [7]	High	Yes	Low	Dynamic frequency selection	Low
Smart agriculture [25]	Medium	Yes	Medium	Channel hopping	High
Industrial IoT [26]	High	Yes	Low	Beamforming	Medium
Connected vehicles [27]	High	Yes	Ultra-low	Coordinated multi-point transmission	High
Healthcare monitoring [28]	High	Yes	Low	Dynamic power control	Low
Smart cities [29]	High	Yes	Low	Interference avoidance	High
Environmental monitoring [30]	Medium	Yes	Low	MIMO	Low
Smart grid management [31]	High	Yes	Low	Cognitive radio	High
Augmented reality [32]	High	No	Ultra-low	Massive MIMO	High
Drones [33]	High	Yes	Ultra-low	Interference alignment	Medium

## 3. Materials and Methods

4G networks are based on LTE technology, which uses a frequency spectrum of around 700 MHz to 2600 MHz. These frequencies are divided into different bands that are used for different purposes, such as voice and data communication. 4G networks use a combination of Frequency Division Duplex (FDD) and Time Division Duplex (TDD) techniques to transmit and receive data. FDD uses separate frequencies for transmitting and receiving data, while TDD uses the same frequency for both.

On the other hand, IoT devices use a variety of technologies and protocols to communicate with each other and with the internet. These devices can operate on different frequency bands, such as 2.4 GHz, 5 GHz, and sub-GHz bands. Some IoT devices use the unlicensed spectrum, which means they can operate on any frequency without needing a license, while others use the licensed spectrum, which requires a license from the relevant regulatory body.

One of the main issues with 4G services and networks is that they can cause interference with IoT devices, especially those operating on the same frequency bands. This interference can cause communication problems and even lead to data loss or corruption. To mitigate this interference, different methods can be used, such as frequency hopping, spread-spectrum techniques, and power control.

Figure 4 summarizes this section, where the main concepts related to the impact of 5G networks on IoT services are observed. We analyze the implementation, the evolution, and the types of wireless technologies that influence current services, the leading 5G technologies and problems, and the coexistence of next-generation networks. We frame interference as the heart of the study of problems in this type of network.

### 3.1. 5G Implementation Requirements

5G is an emerging technology in the telecommunications area. Enabling high-speed connections, offering lower latencies, and ensuring highly scalable connectivity between multiple devices [34], every industry worldwide eagerly waits for mainstream implementation.

5G allows for several technologies related to massive machine-to-machine communications, or IoT, to improve and offer high-speed connectivity between devices, primarily allowing automation in the manufacturing and construction industries. However, IoT has seen massive consumer implementation in homes, malls, and parks, which means this technology has several use cases for ordinary daily consumers.

On the technical side of 5G, it works on a higher spectrum range when compared to traditional wireless technologies. Ranging from 24 to 100 GHz, it provides low-latency communication and high throughput rates but suffers from adverse conditions caused by its high-frequency nature [35]. 5G also operates on two frequency bands: Sub 6 GHz and mmWave (millimeter wave). While mmWave offers faster data transfer rates, it has limited coverage and is easily obstructed. On the other hand, the Sub 6 GHz band, which includes frequencies below 6 GHz, offers wider coverage and can penetrate obstacles such as buildings and walls [36]. This means that 5G can actually work in the Sub 6 GHz band, providing a more reliable and accessible network for users. This is particularly important for rural and suburban areas where building penetration is critical, and for indoor usage where higher frequency bands may not be able to reach. Therefore, despite the hype around mmWave, the Sub 6 GHz band remains a vital part of the 5G network, providing a strong foundation for the future of wireless technology.

To counteract the limitations, 5G has been complemented with new technologies. The most common ones are beamforming, massive MIMO, small cell, mmWave, and network slicing, and new proposals arise everyday to try and get the most out of this generation.

Even though 5G is far from being implemented worldwide, several service providers have been publicly working and sharing their advances related to this technology. Some famous names from the IT world are Ericsson, Verizon, Nokia, AT&T, T-Mobile, Samsung, and Qualcomm. However, undoubtedly, most big telecommunications companies are working on projects related to the new generation of wireless technologies.

Research groups have been focused on the 5G mobile network ecosystem, with institutions such as METIS (Mobile and wireless communications Enablers for Twenty-twenty (2020) Information Society), 5G PPP (5G Infrastructure Public Private Partnership), and NYU New York University Wireless, conducting impactful research related to MIMO transmissions, millimeter waves, and frameworks in the 5G ecosystem [37].

5G networks offer several features that are essential for the Internet of Things (IoT) services, as Figure 5 shows. 5G provides increased bandwidth and lower latency, allowing IoT devices to transmit data quickly and efficiently for real-time communication and response. 5G enables massive machine-type communication (mMTC), which allows a large number of IoT devices to connect to the network simultaneously. Ultra-reliable and low-latency communication (URLLC) is also provided, ensuring quick and reliable data transmission suitable for mission-critical IoT services. Network slicing is another feature of 5G networks, which creates dedicated virtual networks for IoT devices to provide better network performance and security.

Before the implementation of 5G networks, organizations such as IEEE needed to cope with challenges about the implementation of 5G. One of these challenges is about the data rate key dimensions, since the demand of technology services are increasing in the past years. 5G should be a solution in order to satisfy all this demand and provide a great Quality of Service (QoS) [38].

Another factor to consider is the latency, which is responsible for the delay in incoming and outgoing packages on a link, as streaming services such as online games [39], metaverse, and more demand a low latency in order to provide better services.

In modern society, most products use electrical power, including, for example, electric cars such as Tesla. In other words, we are surrounded by electronic devices, and it is essential to focus on developing friendly devices to avoid consuming too much electricity. Some important points about 5G energy consumption need to be mentioned. First, energy efficiency depends on the traffic in the network. According to the article “Energy-efficient 5G for a greener future” [40], when the traffic load is low, a base station can save 98.75% of power, but if the traffic load is high, the power consumption may increase.

Figure 6 describes the requirements for 5G. This generation is the latest generation of cellular networks that is designed to offer significant improvements in performance, capacity, and flexibility compared to previous generations. To achieve these improvements, 5G networks utilize several key technologies. One of these technologies is millimeter wave (mmWave) frequencies, which are higher frequency bands that can provide higher bandwidth and faster data rates. However, these frequencies have short wavelengths and require line-of-sight communication between the transmitter and receiver. Another technology used in 5G networks is massive MIMO (multiple input, multiple output) technology, which uses a large number of antennas to increase the number of spatial streams and improve the efficiency of the wireless channel. This can help to increase the data transfer rates and overall network capacity. Beamforming is also a key technology used in 5G networks, as it uses advanced techniques to focus the radio signal towards a specific device, increasing the signal strength and reducing interference. This technology is crucial for providing reliable and high-speed connectivity in a dense and dynamic environment. Network slicing is another feature of 5G networks that allows for the creation of dedicated virtual networks to meet the specific requirements of different applications. This means that devices can allocated their own network resources and services, providing better network performance and security. Edge computing is also used in 5G networks, which provides real-time processing and analysis of data at the edge of the network. This can help to reduce the latency and improve the performance of time-sensitive applications, such as autonomous vehicles and virtual reality. Low latency is another feature of 5G networks that is important for applications that require real-time communication and response, such as autonomous vehicles and remote surgery. 5G networks aim to achieve a latency of less than 1 millisecond. High reliability is also a critical feature of 5G networks, with features such as network slicing and redundant network paths, ensuring that devices stay connected. This is essential for mission-critical applications that require high availability and low downtime. 5G networks support full duplex communication, which allows for simultaneous transmission and reception of data. This technology can help to improve the efficiency and capacity of the network, enabling higher data transfer rates and reducing latency.

Based on the requirements above, the following are the 5G network deployment requirements:Millimeter wave (mmWave) frequencies for higher bandwidth and faster data rates.Massive MIMO (multiple input, multiple output) technology for increasing spatial streams and improving the wireless channel’s efficiency.Beamforming to focus the radio signal towards a specific device, increasing signal strength and reducing interference.Network slicing for creating dedicated virtual networks to meet the specific requirements of different applications, improving network performance and security.Edge computing for real-time processing and analysis of data at the edge of the network, reducing latency and improving the performance of time-sensitive applications.Low latency for real-time communication and response, aiming to achieve a latency of less than 1 millisecond.High reliability for mission-critical applications, ensuring high availability and low downtime.Full duplex communication for simultaneous transmission and reception of data, improving efficiency, capacity, and reducing latency.

The combination of these technologies in 5G networks can enable a wide range of applications, from IoT services to high-bandwidth applications, such as virtual reality and augmented reality. 5G networks aim to provide the necessary performance metrics to support these applications, enabling the growth of the IoT ecosystem and a new era of connectivity.

### 3.2. Evolution of Wireless Networks

Wireless networks [41] have an important role in our society, because they allow us to keep connected in a network without cables, being the bridge to create useful tools. For instance, 3GPP provides basic data services such as voice and messaging capabilities, with 4G we can make video calls, and use IP services, and with 5G we can get more benefits, such as low latency, more bandwidth, and other interesting features [42].

### 3.3. Evolution of IoT

The evolution of the internet allowed remote connections between machines, and the protocol standardization used to transmit the information, such as TCP/IP, enabled researchers to take advantage of the internet and allowed to creation of new protocols and ways to transmit the data [43]. Since then, new technologies has arrived in our lives, which is the case of IoT. In the last few years, new technologies and protocols arose to connect and transmit data between IoT devices and networks, each providing specific features to perform the applications. In the following statements, we mention some protocols to communicate data. Each protocol can provide better communication depending on the application. For example, LoRaWAN is used in long-range communication, allowing to transmit of data through large distances even with obstacles between the link and using low power consumption. Zigbee is a protocol commonly used in smart homes, avoiding inferences by routers, electronic devices, and so on.

The evolution of IoT technologies is having a significant impact on 5G networks. As more and more devices are connected to the internet, there is a growing need for faster, more reliable connectivity. This is where 5G networks come in, providing higher speeds, lower latency, and greater capacity than previous wireless technologies. The proliferation of IoT devices also creates new opportunities for 5G, as the technology is able to support the massive amounts of data generated by these devices. However, this also poses challenges for 5G networks, such as the need to handle large amounts of traffic from a variety of devices with different requirements. As a result, the evolution of IoT technologies is driving innovation in 5G networks, as providers look for ways to meet the demands of this growing ecosystem.

The implementation of 5G networks is expected to have a significant impact on the evolution of IoT services. One of the most significant changes is the increase in the speed of data transfer. 5G networks have much faster data transfer speeds, lower latency, and higher capacity than previous generations of mobile networks [44]. This means that IoT devices can transmit and receive data much more quickly, which enables more real-time data processing and analysis. The improved responsiveness of IoT services will open up opportunities for new applications that require near-instantaneous data processing, such as remote surgery, autonomous vehicles, and industrial automation.

Another important change brought about by the implementation of 5G networks is the increased scalability of IoT services. 5G networks have a higher device density, which means that they can support a greater number of IoT devices per unit area. This will help to increase the scalability of IoT services and allow for the deployment of more complex and sophisticated IoT solutions.

Furthermore, 5G networks are expected to improve the reliability of IoT services. 5G networks have more robust error correction capabilities and redundancy features, which can increase the reliability of IoT services. This is particularly important for mission-critical applications, such as remote monitoring of infrastructure or medical devices [45]. Improved reliability will also be beneficial in industries that require highly reliable communication networks, such as manufacturing and energy.

The enhanced security features of 5G networks will also impact the evolution of IoT services. 5G networks offer improved security features, such as network slicing, which allows for the creation of isolated virtual networks for different IoT applications. This can help to prevent unauthorized access and ensure the security and privacy of IoT data. With the increasing number of IoT devices, the importance of security and privacy is critical and 5G networks can provide a more secure environment for the data transmitted.

5G networks will help to reduce the power consumption of IoT devices. 5G networks are designed to be more energy efficient than previous generations of mobile networks [46]. This can help to reduce the power consumption of IoT devices, prolonging their battery life and reducing their environmental impact. This is particularly important for IoT devices that are difficult or expensive to replace or recharge, such as sensors deployed in remote or inaccessible locations. Figure 7 describes the main features related to 5G networks and IoT services. The performance metrics associated with energy consumption are mentioned here too.

### 3.4. Wireless Communication Technologies for IoT and Cloud-Based Solutions

This section covers various wireless communication protocols used in IoT devices, including Wi-Fi AdHoc, Zigbee, Z-Wave, LoRaWAN, and SigFox. It also touches on the use of cloud computing for IoT applications, such as data storage, analytics, and remote device management. This section could provide a comprehensive overview of wireless communication technologies used in IoT devices and their integration with cloud-based solutions.

Wi-Fi AdHoc: With the standard of IEEE 802.11, Wi-Fi technology turned into the first technology to create devices connected to the network. Allowing to create news architectures such as Wi-Fi AdHoc is a decentralized type of wireless network because each node participates in routing by forwarding data to other nodes [47]. These nodes can be IoT devices and are very helpful in applications where it is needed to have many devices connected.Zigbee: The most popular industry wireless mesh networking standard for connecting sensors, instrumentation, and control systems. Zigbee implements communication in a personal wireless area network, providing low power consumption and interoperating multi-vendor, commonly used in home automation, low-power consumption sensors, HVAC (Heating, Ventilation, and Air Conditioning) control, etc. [48].Z-Wave: A wireless protocol evolved by Zensys and confirmed by the Z-Wave Alliance for automation apparatuses for home and commercial environments. This protocol allows transmitting short messages with minimum noise and uses a Mesh network configuration [49].LoRaWAN: A low-power, wide-area networking protocol designed to connect battery wirelessly operated ‘things’ to the internet in regional, national, or global networks. It targets IoT requirements, such as bi-directional communication, end-to-end security, mobility, and localization services. According to work cited in [50], LoRa has the most features in terms of IoT, such as low power consumption, long-range communication, etc. Furthermore, the paper tested communication in urban and forest areas, showing that LoRaWAN can transmit data up to 2.1 km in urban areas.SigFox: SigFox is a network operator dedicated to the Internet of Things. The SigFox network uses the ultra-narrow band, allowing devices to communicate with low power on a wide area [51].Cloud Computing: Cloud computing is a term used to describe both a platform and a type of application. One of the essential features of cloud computing is the capability to assign dynamic resources to the network, being an important key to providing scalable solutions and avoiding high costs. The interference plays an important role when connecting devices since the signal quality decrease, which means the modulation decreases and the bits per error increases. That is a problem if we are trying to offer large bandwidths and low latency in each data transmission [52]. We must consider different factors to provide a great QoS, like the weather, buildings, hardware, software resources, etc. In the IoT context, the buildings and distances create the main interferences. For that reason, technologies such as Zigbee, SigFox, LoRaWAN, and Z-wave play a vital role in connecting devices.WiGig: WiGig, also known as 802.11ay, is a wireless communication technology that operates on the 60 GHz frequency band [53]. It was developed as an extension of the Wi-Fi standard to provide high-speed, short-range wireless communication, primarily for applications that require high bandwidth, such as virtual reality, high-definition video streaming, and gaming. WiGig supports multi-gigabit data transfer rates, with theoretical speeds of up to 176 Gbps, which is much faster than the previous Wi-Fi standards [54]. It achieves this speed through the use of wider bandwidth and advanced modulation techniques, such as Quadrature Amplitude Modulation (QAM) and Orthogonal Frequency Division Multiplexing (OFDM). Another notable feature of WiGig is its low latency, making it ideal for applications that require real-time data transfer, such as gaming and virtual reality [55]. It also supports multiple-input, multiple-output technology, which enables multiple antennas to transmit and receive data simultaneously, improving the overall performance and efficiency of the network. In the context of 5G networks, WiGig can be used as a complementary technology to provide high-speed local area network (LAN) connections for mobile devices and IoT devices. The 60 GHz frequency band has a limited range, but it can support high data rates over short distances, making it suitable for applications, such as augmented and virtual reality (AR/VR), wireless HD video streaming, and cloud gaming [56]. In addition, WiGig can be used as a backhaul technology for small cells in 5G networks, enabling high-speed data transfers between small cells and the core network. This can help improve the performance and capacity of 5G networks, especially in densely populated urban areas where there is high demand for data services. Regarding IoT services, WiGig can enable high-speed local area connections between IoT devices, allowing them to share data quickly and efficiently. This can be especially useful for applications, such as smart homes, where multiple IoT devices need to communicate with each other in real time.

### 3.5. 5G Applications

5G benefits are not utilized solely by the most prominent IT corporations worldwide. In the modern era, we have found several exciting applications where 5G’s high speeds, excellent reliability, and energy efficiency come into play and provide a better user experience.

Entertainment services: Video-on-demand services are currently one of the most utilized services on the internet. These services demand high-speed connections, and with rising trends to utilize higher resolution devices, 5G plays a vital role in providing optimal user experience so that they can consume their content without interruption.General mobile networks: Due to the COVID-19 pandemic in recent years, teleworking has seen an immense rise in all sectors globally. This means workers must be able to respond to video or voice calls at any given time, requiring improved downlink and uplink speeds. These requirements, complemented with the higher reliability aspects of 5G, mean that this implementation will improve communications in any given context, especially for work-related tasks.Internet of Things: IoT is one of the trendiest topics around the electronics ecosystem, due to its nature to provide automation to simple or very complex topics. Even though most IoT devices currently utilize 3G or 4G-LTE technologies due to their low requirements for data connectivity, a new generation of IoT devices requires higher throughput rates. These requirements are on the limits of the current generation of wireless technologies, which makes 5G an interesting contestant to solve these requirements. The most popular IoT applications today involve Smart Homes, industries, or farming, which generally require low amounts of wireless capabilities because the devices have low microprocessing power due to the nature of the technology itself. However, new trends, such as Smart Cities or IoV (Internet of Vehicles), require much greater throughput to function correctly and offer an optimal user experience. These types of solutions require capabilities that are only offered by 5G. IoT data are visioned to increase in data provided per area by 1000 times [57] which means IoT applications will be part of our everyday lives. The current data are provided mainly by sensors, but more complex devices will mean that data will be gathered from additional sources. This exponential growth in data consumption will also need to be stored in scalable data storages, and this is where cloud computing comes in. Some of the most famous architectural trends of IoT devices follow three principles:
Hardware: All sensor nodes that gather data, their communication methods, and the hardware interface with the user.Middleware: The layer in charge of storing and analyzing the data and monitoring the devices.Presentation layer: Commonly called the front end, this presents visualization tools better to understand our devices’ current state and behavior.5G technologies can impact the hardware and middleware layer. As the technology is more energy efficient, the devices do not need massive antennas, which can lead to increased consumption. At the middleware layer, as devices can communicate more data in a given period, this layer will benefit from extra data to improve any statistical or machine learning model.

### 3.6. 5G Technologies

The purpose of this section is to describe some of the most popular technologies implemented with 5G.

Figure 8 shows characteristics and technical specifications of 5G. 5G networks offer several key characteristics and technical specifications that make them ideal for a wide range of applications. 5G provides significantly increased bandwidth, enabling faster data transfer rates for high-speed applications, such as virtual and augmented reality. It also offers lower latency, making it suitable for real-time communication and response applications, such as autonomous vehicles and remote surgery. 5G is highly reliable, with features such as network slicing and redundant network paths, ensuring devices stay connected. 5G enables massive machine-type communication (mMTC) and ultra-reliable and low-latency communication (URLLC), making it suitable for mission-critical applications, such as industrial automation. Additionally, 5G offers network slicing, beamforming, and uses millimeter wave (mmWave) frequencies for even higher bandwidth and faster data rates. Overall, 5G networks offer significant improvements in bandwidth, latency, reliability, and flexibility, making them well-suited for a wide range of applications.

Massive MIMO: This technology is responsible for sending and receiving multiple signals simultaneously, utilizing the same radio channel. While other technologies, such as Wi-Fi or 4G-LTE, have utilized this technology, massive MIMO performs best when paired with 5G technologies. This technology uses extra antennas to move energy into smaller regions of space, which means spectral efficiency and coverage are improved [58].NOMA: Non-Orthogonal Multiple Access: A radio access technology that plays a vital role in 5G applications [59]. This technology offers several benefits, such as low latency and massive high-speed connectivity. Code domain NOMA is commonly paired with mMIMO, drastically improving spectral efficiency [60]. Power domain NOMA is commonly utilized with MIMO, beamforming, and even cooperative communications, being one of the most flexible technologies utilized in 5G implementations.Millimeter Wave: This technology uses a frequency band between 30 GHz and 300 GHz and derives its name from the 1 to 10 mm waves utilized by the technology. Utilized commonly in radar applications, this technology is being paired with 5G to improve spectrum bandwidth and increase spectrum utilizations. The main benefit of pairing this technology with 5G is the spectrum freedom linked to mmWave. Standard technologies, such as GPS, 4G, and satellite connections, utilize the 1 GHz to 6 GHz spectrum, which is becoming very crowded [61]. Because mmWave is new and has a massive spectral range, 5G provides an improved user experience through this combination.Machine Learning Techniques: Supervised and unsupervised models are being implemented in 5G technologies to improve overall network capacities, predict energy consumption, and optimize tracking technologies such as beamforming. In the supervised category, some 5G networks utilize Linear Regression Algorithms to predict the scheduling of nodes [62]. Other supervised models utilize Deep Neural Networks to predict beamforming vectors. Then, unsupervised learning models are used to improve handover selection and reduce interruption of services, as well as they can reduce latency by clustering fog nodes.Unmanned Aerial Vehicles (UAV): Being the most innovative proposal, current 5G researchers are utilizing UAVs to improve network coverage. These UAVs will assist the terrestrial network by serving as beacons. The high altitude of these planes could solve many interference problems and even replace entirely terrestrial cellular networks [63]. UAVs, commonly known as drones, have become increasingly popular for both commercial and personal use. With the advent of 5G networks and IoT, UAVs can now be equipped with a wide range of sensors and devices that can transmit real-time data to ground stations for analysis and decision making [64]. There are several potential implementations of UAVs in the context of 5G and IoT. One such implementation is in the area of precision agriculture. UAVs can be used to gather data on crop growth, soil conditions, and other factors that affect agricultural production. These data can be transmitted in real time to a ground station for analysis and used to optimize planting, fertilization, and irrigation schedules [65]. 5G networks and IoT sensors can provide the necessary bandwidth and low latency for this type of application. Another potential application of UAVs in the context of 5G and IoT is in industrial inspection and maintenance. UAVs equipped with cameras and other sensors can be used to inspect and monitor equipment and infrastructure such as power lines and wind turbines. Real-time data transmission via 5G networks can enable remote monitoring and control of these systems, improving their reliability and reducing maintenance costs [66]. UAVs can also be used for emergency response and disaster management. In the event of a natural disaster, such as a hurricane or earthquake, UAVs can be deployed to assess damage and provide real-time information to first responders. The data collected can be transmitted via 5G networks to emergency management centers for analysis and decision making. UAVs can be used for surveillance and security purposes. In public safety applications, UAVs can be used to monitor crowd movements and gather intelligence on potential threats. In private security applications, UAVs can be used to patrol and monitor facilities for intruders or other security threats. The combination of UAVs with 5G networks and IoT sensors can enable a wide range of applications in various industries. With the potential to improve efficiency, reduce costs, and enhance safety, it is likely that we will see an increase in the adoption of UAVs in the coming years.

### 3.7. 5G Problems

5G offers excellent improvements over last-generation mobile communication technologies. However, many problems related to technological complications, security and privacy implications, and even social implications have been discovered.

Technical complications are related primarily to interference. 5G as a technology has complications related to interference, being as sensible as being dramatically affected by mild rain in urbanized areas [67]. Even though the technologies mentioned previously help mitigate these problems, 5G still needs to be ready to be used on a massive scale. Tests performed in several countries utilizing small cells show the need for more extensive architectures to offer full coverage and optimal user experience [68]. This means costly architectures will lead providers to focus mainly on urban areas, leaving rural areas unattended. This leads to the following obstacles, which are ethical and social implications. One of the main advantages of 5G is the ability to offer improved connections to people in poor conditions. However, right now, it is only used as a marketing stunt because 5G architectures are considerably expensive and cannot offer coverage in rural areas. As 5G equipment and technology become more widely adopted in the following years, we expect big strides in this department.

Another ethical and social implication is IoV technologies enabled by 5G. As we see more autonomous driving cars, and 5G will enable more precise driving for these cars, as well as broad adoption of them, we are encountering huge moral dilemmas in cases where car accidents are caused by autonomous vehicles. Security implications arise from the trend of connecting vehicles to the internet, which means people can be kidnapped remotely by their car if hackers gain control of their vehicle via the web.

The main problems mentioned and explained above are listed below:Technical complications related to interference, including sensitivity to mild rain in urban areas.Need for extensive and costly architectures to offer full coverage and optimal user experience, which may lead providers to focus mainly on urban areas, leaving rural areas unattended.Ethical and social implications related to the inability to offer improved connections to people in poor conditions due to the high cost of 5G architectures and the lack of coverage in rural areas.Security implications arising from the trend of connecting vehicles to the internet, which could result in remote kidnappings by hackers.

Table 2 compares the problems presented by 5G and IoT services across several key areas, namely security, latency, interference, cost, and compatibility. In terms of security, 5G networks are vulnerable to a range of threats, such as DDoS attacks, identity theft, and man-in-the-middle attacks, while IoT devices can suffer from security breaches due to weak encryption, passwords, and outdated firmware. Regarding latency, 5G networks have lower latency, which can be a problem for certain IoT applications that require real-time response, while IoT services may suffer from latency due to network congestion, distance from the server, and the number of devices connected to the network. Interference is another factor that can affect both 5G and IoT services, with both being susceptible to interference from other wireless devices and environmental factors. Cost is also a concern, as the high cost of 5G infrastructure and services may limit its usefulness for many IoT applications, while IoT services may also be expensive to deploy and maintain, particularly if they require high bandwidth or specialized hardware. Finally, compatibility is a potential issue for both 5G and IoT services, with some IoT devices not being compatible with 5G networks, and IoT services limited by compatibility issues with proprietary hardware or software.

### 3.8. Coexistence of 5G Networks and IoT Applications

The coexistence of 5G networks and IoT applications has the potential to revolutionize several industries by providing ultra-reliable and low-latency communication, massive machine-type communications, and network slicing for multiple services and applications [18]. One novel idea is the use of 5G networks for industrial automation and autonomous vehicles, where ultra-reliable and low-latency communication is critical for ensuring the safety and reliability of the system. Another innovative application is the use of 5G networks for vehicle-to-everything communication, where autonomous vehicles can exchange information with other vehicles and infrastructure to improve traffic management and safety [69]. The use of 5G networks for smart cities and agriculture can also be highly beneficial, as it can enable massive machine-type communication to support a large number of IoT devices, such as sensors and actuators, and facilitate data collection, analysis, and decision making. Additionally, the use of network slicing can enable multiple services and applications to coexist on the same infrastructure while maintaining their unique requirements for security, reliability, and latency. However, the coexistence of 5G networks and IoT applications also presents several challenges, such as security, interference, compatibility, and cost [35]. For example, security is critical for protecting IoT devices and networks from cyber attacks, while interference from other wireless devices and environmental factors can affect both 5G and IoT services. Moreover, compatibility issues can arise when IoT devices are not compatible with 5G networks, and the high cost of deploying and maintaining 5G infrastructure and IoT services can limit their widespread adoption. Overall, the coexistence of 5G networks and IoT applications presents both opportunities and challenges and requires careful consideration of their technical requirements, security, and compatibility to realize their full potential.

Table 3 provides a summary of the coexistence of various 5G services and networks with IoT applications. It highlights the wireless technology, algorithms used, and coexistence of services with 4G and Wi-Fi. The table includes several 5G services, such as 5G NR, 5G-V2X, 5G mMTC, 5G-UHD, 5G-IoT, 5G-eMBB, 5G-URLLC, 5G-mIoT, 5G-gNB, and 5G-Slicing, and describes their potential applications and system description. The 5G services offer different benefits and are designed for specific applications, such as ultra-reliable and low-latency communications for industrial automation, vehicle-to-everything communications for traffic management and safety, massive machine-type communications for smart cities and agriculture, ultra-high-definition video communications and virtual reality, Internet of Things communications for smart homes and wearables, enhanced mobile broadband for high-speed data transfer and streaming, ultra-reliable and low-latency communications for mission-critical applications, massive IoT communications for smart cities and agriculture, gigabit-class communications for high-speed data transfer and streaming, and network slicing for multiple services and applications. The table provides a useful overview of the coexistence of 5G and IoT services and can be helpful in understanding the potential benefits and challenges of deploying these technologies together.

### 3.9. Interference and Network Optimization Difficulties

Nowadays, with the development of new technologies, networks face different challenges. Issues such as coverage, latency, availability, and accessibility, among others, appear. Moreover, taking into account a more profound or more centralized point to the characteristics of the network, issues such as latency, interference, spectrum use according to the technology being used, and others appear.

Interference in 5G networks can occur due to a variety of factors such as environmental conditions, radio frequency congestion, and the deployment of too many access points in a limited area. These issues can cause signal loss or degradation, resulting in poor network performance and reduced data transfer rates. Interference in 5G networks can have a significant impact on IoT devices that rely on a stable and reliable connection to operate effectively.

IoT devices are designed to transmit small amounts of data over long periods, often using low power and low bandwidth networks. The interference in 5G networks can cause delays or interruptions in the transmission of data between IoT devices and the cloud-based systems they rely on. These disruptions can impact critical functions, such as real-time monitoring of environmental conditions, traffic management, and energy consumption. In addition, interference in 5G networks can result in increased latency, which can make it difficult for IoT devices to communicate with each other in a timely manner.

Another critical aspect of interference in 5G networks is the potential for security breaches. As the number of connected devices increases, the risk of cyber attacks also grows. Interference in 5G networks can make it easier for hackers to gain unauthorized access to IoT devices and the data they transmit. This can have serious consequences, particularly for devices that are used in critical infrastructure, such as healthcare systems, power grids, and transportation networks.

To mitigate the impact of interference in 5G networks on IoT devices, it is essential to ensure that the network is properly configured and managed. This includes deploying access points strategically to minimize congestion, using directional antennas to focus signals and reduce interference, and implementing security measures to prevent unauthorized access. In addition, new technologies such as edge computing and network slicing can help to reduce latency and improve the reliability of IoT device communication.

Table 4 outlines some of the main types of interference that can impact 5G networks and the IoT. Interference can occur due to a variety of factors, such as noise, multipath, co-channel interference, interference from other devices or wireless networks, and jamming. One of the key aspects of network performance that interference can affect is coverage. Interference can reduce the coverage of a wireless network by increasing the signal-to-noise ratio (SNR) threshold required for reliable communication. When there is noise, multipath, or interference from other wireless networks present, signals may not reach as far or penetrate as deeply into buildings, resulting in reduced overall coverage. Another aspect that can be affected by interference is latency. Interference can increase latency by introducing delays in signal transmission or reception. For instance, multipath interference can cause signals to arrive at a receiver at slightly different times, leading to signal distortion and increased latency. This delay can be particularly problematic for real-time applications, such as gaming or video conferencing. Interference can also impact the availability of a wireless network. When there is co-channel interference or interference from other devices, it can cause collisions or channel saturation, leading to reduced availability. This can result in packet loss or connection drops, making it difficult for users to access the network and resulting in a poor user experience. Access to a wireless network can also be affected by interference. Interference can make it harder for devices to connect to a network, leading to connection failures or reduced bandwidth. This can be particularly problematic in areas with high device density, such as urban environments. Interference can also affect modulation and coding. When there is interference present, the quality of the signal can degrade, making it more difficult for the receiver to demodulate and decode the signal. This can result in errors and reduce the overall throughput of the network.

#### 3.9.1. Interference

One of the issues highlighted by the development of 5G is interference. It should be remembered that interference, by definition, is when two electromagnetic waves overlap. This overlapping causes communication failures from where the wave is sent to where it is received. There are two types of interference: constructive and destructive. Constructive interference occurs when the waves overlap or collide in phase, and this causes the resulting wave to have a greater amplitude. Furthermore, destructive interference is when there is an out-of-phase overlap, where the resultant of this wave will be of lower amplitude [80]. Some networks may be more susceptible to interference depending on the type of wave they handle, i.e., the frequency used in the electromagnetic spectrum. RF can give this interference, whereas in 5G networks, millimeter bands and below 6 GHz bands are used. In addition, it must be taken into account that these interferences will continue to grow with the development of technology.

Figure 9 covers various types of interference in wireless communication systems in 5G. Adjacent channel interference occurs when adjacent frequencies or channels overlap due to imperfections in filters or non-linearity in amplifiers. Inter-cell interference happens when two users from neighboring cells try to use the same frequency band simultaneously due to resource limitation within the network. Intra-channel and inter-channel interference occur due to physical proximity of devices and low power within the macro-cell network, respectively. Inter-symbol interference results from signal distortion due to phase or amplitude dispersion in the channel. Inter-carrier occurs due to frequency offsets between sub-carriers. Inter-numerology interference arises due to non-orthogonality in the system, causing difficulties for symbol alignment. Cross-link interference occurs when signals are transmitted to neighboring cells in different directions simultaneously. Inter-beam interference happens due to the use of multi-beam antenna systems, causing interference from adjacent beams. Multi-user interference happens when several users try to transmit their information requests at the same time, and MU-MIMO is used to increase the capacity and performance of wireless broadcasting systems.

We must consider that with the evolution of higher data rates, we are looking for the implementation in a frequency spectrum that is becoming more saturated daily. The development in these frequencies makes the operator reuse the same frequencies for neighboring cells to save spectrum. This development causes high interference between cells [81]. In addition to these, other interferences also occur within the networks. The classification of these interferences is as follows:Adjacent channel interference: Adjacent channel interference is a problem that occurs in many devices and many frequency ranges. This occurs when an adjacent frequency or the adjacent channel of the one used by our device overlaps. This problem occurs in 5G, as in other mobile technologies. This type of interference is mainly caused, as stated in [82], by an imperfection in the filters, where they cannot filter the desired signal correctly. These imperfections result in nearby frequencies passing into the passband. In turn, this occurs because the amplifiers are not linear.Intra-cell and inter-cell interference: Inter-cell interference is one of the most significant causes of network performance degradation. This occurs when two users from neighboring cells attempt to use the same frequency band simultaneously. This occurs, as said in [83], because of resource limitation within the network, due to the frequency reuse factor. In the same way, inter-cell interference is identified when a Base Station connects directly in the close range of another BS, and that is when there is a simultaneous transmission in that reuse. A distortion appears by the interference that gives the user and other equipment in the same cell.Intra-Channel and Inter-Channel interference: Inter-channel interference usually occurs when there is a low power within the macro-cell network. Then, when communicating, the information is forwarded to the nearest base station. When the transmission is made, it is completed through fast switching and restricts delays and propagation losses. However, this produces intra-channel interference that affects the valuable signal. On the other hand, inter-channel interference occurs due to the physical proximity of the devices when two separate frequency bands cause interference with each other. As these devices operate in close range, the transmitter of a high-power signal interferes with the receiver of a weak signal. It is important to emphasize that 5G IoT networks or devices with channels in the MHz range are susceptible to this type of interference due to the proximity and number of nearby devices. One way to mitigate this type of interference is through spatial modulation via MIMO [84].Inter-Symbol Interference: Inter-symbol interference occurs when one or more symbols interfere with other symbols. It is caused by phase or amplitude dispersion in the channel, resulting in signal distortion. This can be seen in OFDMA, where multipath propagation occurs. One study [85] exposed how this type of interference can be an excellent challenge for network systems and should be sought to improve the efficiency of the bandwidth while seeking alternatives in the modulation. This is in order to counteract this type of interference.Inter-numerology interference: Multiple numerology is a model to provide flexibility for devices in different services. For these numerologies, 15, 30, 60, 120, and 240 kHz channels are used. They aim to improve performance and significant bandwidth. As stated in [86], the authors introduce a non-orthogonality in the system, causing difficulties for symbol alignment in the time domain. When sampling at the same frequency, numerology tends to align differently, making synchronization within the frame difficult. This is known as interference between numerologies.Cross-Link Interference: This interference occurs when signals are transmitted to neighboring cells in different directions simultaneously, either in time frequency or arbitrarily overlapping resources. As mentioned in [87], there are different types of interference and different ways in which it occurs. For example, the Base Station receives interference from user equipment devices in adjacent cells, or a downlink user equipment receives interference from a second database.Inter Beam Interference: With the high demand for technological services, the aim has been to improve spectral efficiency and network performance. In trade-offs for networks such as 5G, where the quality and capacity must be high, the solution of incorporating a multi-beam antenna system such as mMIMO was sought, as explored in detail in [88]. This technique identifies the best route to providing optimal performance to a user. This approach helps compensate for transmission attenuation losses, especially in millimeter-wave communication. In this case, the base station generates multiple narrow beams of mainly RF energy in all directions of the coverage area. This causes a spatial division of multiple beams, introducing interference. Adjacent beams cause this interference from the same cell or a neighboring cell.Multi-user Interference: Multi-user interference is due to the industry’s quest for higher data rates in applications and the dramatic increase in subscribers to wireless communications [89]. The techniques used for this type of technology are given by MU-MIMO, a 5G generation technique that helps increase the capacity and performance of wireless broadcasting systems. Multi-user interference occurs when several users try to transmit their information requests at the same time.

#### 3.9.2. Interference in 5G

The different classifications of interference have been explored. Now it is sought as part of the state of the art to understand interference within 5G. The enrichment of interference in technologies such as Hetnets, IoT, D2D, Relay Node, and New Radio 5G will be sought.

Interference in Hetnets: Hetnets are 5G heterogeneous networks. They aim to provide wireless coverage for mobile subscribers and indoor and outdoor applications. Hetnets are multi-tier network systems that deploy small cells in populated areas. These cells are characterized by having a short range and low power consumption. This network comprises access points, which allow high density, improvements in network flexibility, and so on [90]. Due to the infrastructure and locations for deploying this type of network, they are prone to different types of interference, such as intra-cell, inter-cell, and adjacency channels. This type of problem is due to its poor systematization and organization in its network design. There are types of interference within heterogeneous networks due to their infrastructure or grouping depending on the need to be met. The first interference that appears is Co-tier interference. This type of interference appears mostly in femtocells, where there is much demand for higher data rates. This environment allows coverage, such as low-power radio access points, giving various services at home [91]. This type of interference is observed when multiple users reside in the same network tier, where transmission occurs over adjacent cells within the femtocell. Similar to co-tier, cross-tier interference can appear. The difference is that the co-tier is the inter-cell interference, and the cross-tier is the interference between the femtocell and macro-cells, considering that the femtocell is inside the macro-cell [92]. Last but not least, in [93], channel control interference says that one of the most critical factors for channel control is the physical control format indicator channel. This channel carries scheduling and synchronization information for the uplink and downlink link data channels. As the transport is by physical means, this will induce interference.Interference in D2D: With the introduction of 5G, the best wireless systems have constantly sought a solution that allows the best quality of communication. Device-to-device networks are one of the candidates to be the future of 5G networks. Direct contact between two mobiles increases efficiency in the spectrum. However, it always brings some challenges, in this case dealing with interference. As discussed in the description of Hetnets above, these use a high connectivity capacity thanks to their structure with macro-cell and femtocell. They allow good performance [94]. With the introduction of D2D, a cellular network is sought that significantly improves spectral efficiency and performance. However, some challenges are the need for more security that this type of technology presents and interference. On the interference side, they appear related to inter-cell interference. Furthermore, intra-cell may be related to adjacent frequency. Furthermore, we must remember that there are types of interference typical of D2D nodes, D2D-to-CU interference, and inter-D2D node interference, among others.Interference in IoT and Smart Cities: We cannot limit our minds to just one application when discussing the Internet of things. However, everything from D2D devices and V2X to smart homes or buildings is part of it. The conception of IoT with 5G has evolved in large and small ecosystems. IoT in some applications is used over the unlicensed ISM band, which is used for various physical devices to properly leverage the spectrum to adhere to the conditions and regulations of short radio communication. The Internet of Things has been seen as the other great leap in the evolution of the Internet. With this in mind, developing smart cities will help create a more sustainable and cost-efficient ecosystem [95]. The combination of technologies such as 5G, IoT, and others will enable big data, offering complex services to the community, adding members in smart cities, and ensuring compatibility. However, it should be noted that interference must be characterized. When talking about smart cities using Wi-Fi because of the use of the millimeter band, physical obstacles such as walls are things to keep in mind, in addition to channel overlapping and inter-carrier interference [96].

#### 3.9.3. Optimization Challenges in 5G Networks

5G is presented as a developed environment with an ecosystem where the boundaries are complete cities. 5G technologies promise high-speed and low-latency data transmission using the millimeter band. Although this frequency band allows these advantages, it only allows short distances. By allowing such short distances, problems such as interference become apparent. The optimization of this type of network should always be used to get the best out of it. In 5G, techniques such as MIMO and beamforming are used to reduce interference or signal degradation [90]. It should be noted that optimizations for 5G networks include different architectures that are part of this type of network. These architectures can be divided into non-standalone and standalone.

Optimizing these architectures, depending on their niche, can be profitable, as the communication varies. Communication can be more accessible or restricted depending on the architecture. At the same time, different modulations can be obtained to enhance data transmission, transmission speed, and latency. In the same way, with 5G, most of the architectures allow the incorporation of MIMO. Thanks to beamforming management, it is possible to optimize the use of frequencies for transmission so that some technologies can send information in the mmWaves range.

From interference in the case of NR, other interferences negatively affect the data transmission in this architecture due to the nature of this architecture. These are inter-carrier interference and phase interference. This is because it uses the mmWave frequency range. In turn, another interference that damages NR is numerology interference [97].

The evolution of mobile networks from 3G to 4G and now to 5G has significantly impacted the services and Internet of Things devices. Each new generation has provided faster data transfer speeds, increased bandwidth, and improved reliability, enabling the emergence of new services and applications that were previously not possible on mobile devices. As 5G networks become more widely adopted, they will continue to unlock new opportunities for innovation, transforming the way we interact with our devices and the world around us.

Table 5 shows the impact of each mobile generation starting with 3G and explains the impact in an IoT context.

New technologies and recent research about wireless communication have created new ways to transmit and receive data. With the evolution of technology such as 5G, new features arrived to provide/perform the ways to send and receive data. IoT has shown more benefits, helping with other technologies such as machine learning, cloud computing, and others. Since the data can be processed, each device forms a swarm intelligence that allows automatization and avoids wasting resources in the industry.

Another factor in analyzing is the interference with the channels, since our houses, offices, and even natural places have electronic devices that typically use the 2.4 GHz band, which causes interference in this band. Therefore, 5G incorporates new bands to avoid the interference caused by this massive consumption of 2.4 GHz. We need to compare the features of 3G, 4G, and 5G to understand why 5G is an essential key in smart cities. The main technical characteristics of these three generations of technologies can be seen in Table 6.

Table 7 presents the expectations for the use of 5G technology in the IoT. The first feature mentioned is bandwidth. Bandwidth refers to the amount of data that can be transmitted over a network or a communication channel in a given amount of time. In the context of 5G, higher bandwidth means that more data can be transmitted over the airwaves in a shorter period of time, which can result in faster and more reliable 5G transmissions. Another feature highlighted is artificial intelligence, which can analyze all the data generated by the IoT devices to take accurate decisions. With the use of AI, IoT devices can identify patterns, make predictions, and take actions based on the data gathered. This feature is essential to support IoT applications that require real-time decision making, such as smart cities and intelligent transportation systems. The third feature is real-time monitoring and management, which means the capability to monitor and manage electronic devices. This feature allows IoT devices to be monitored and managed from a central location, which can be very useful for large-scale applications, such as industrial automation or smart grid management. Swarm intelligence is another feature mentioned, which means that each IoT device can be a node and work with other nodes to create a swarm intelligence. This feature enables IoT devices to work collaboratively and efficiently, providing better performance and reliability in complex applications. Quality of service (QoS) is also listed as a significant feature that 5G can provide in IoT connections. With 5G, IoT devices can experience better connectivity, lower latency, and higher throughput, which can improve the overall QoS and performance of IoT applications. In the context of IoT connections, QoS refers to the ability of the network to provide reliable and high-performance connectivity to IoT devices. 5G is a promising technology for IoT connections, as it can provide several features that can improve the overall QoS and performance of IoT applications. 5G is designed to provide a significantly improved QoS compared to previous generations of mobile networks. This is achieved through a combination of advanced technologies and features that are built into the 5G standard. One of the key features of 5G that can contribute to a better QoS is the use of advanced radio access technologies. These technologies, such as massive MIMO and beamforming, allow for more efficient use of the radio spectrum and better signal quality. This means that 5G can provide better connectivity and signal strength, which can improve the reliability and availability of the network. Another important feature of 5G that can contribute to a better QoS is the use of network slicing. Network slicing allows the network to be divided into multiple virtual networks, each tailored to specific use cases and requirements. This means that different applications and services can be given different levels of priority, bandwidth, and latency requirements, depending on their needs. This can improve the QoS for each individual application and user. In addition, 5G can provide lower latency, higher throughput, and better support for massive IoT deployments, all of which can contribute to a better QoS. Lower latency means that there is less delay between the transmission of data and its reception, which can enable real-time communication and faster response times. Higher throughput means that more data can be transmitted over the network in a given time, which can improve the performance of applications that require high-bandwidth data transfer. Finally, better support for massive IoT deployments means that the network can handle a large number of connected devices and data traffic, without compromising the QoS for individual devices or applications.

Low latency is another feature that 5G can offer, reducing the delay between data transmission and reception. This feature is particularly important for applications that require immediate decision making, such as autonomous vehicles or remote surgery. Cloud computing is mentioned as a feature that 5G can provide in IoT connections. With 5G, IoT devices can connect to the cloud more efficiently, and each device can use all the resources of the cloud, such as storage and processing power. This feature can significantly improve the performance and scalability of IoT applications, especially those that require large amounts of data processing and storage.

Low latency is particularly important for applications that require immediate decision making, such as autonomous vehicles or remote surgery. With 5G, the latency can be reduced to as low as one millisecond, which can enable real-time communication and faster response times. Furthermore, 5G can provide higher throughput, which refers to the amount of data that can be transmitted over a network in a given time. With higher throughput, IoT devices can transmit and receive larger amounts of data, which can improve the performance of IoT applications that require high-bandwidth data transfer, such as video streaming or real-time sensor data. Cloud computing is another feature that 5G can offer for IoT connections. With 5G, IoT devices can connect to the cloud more efficiently, and each device can use all the resources of the cloud, such as storage and processing power. This feature can significantly improve the performance and scalability of IoT applications, especially those that require large amounts of data processing and storage.

Since smart cities use IoT devices to provide a solution in our daily routine, 5G is essential in providing a more stable, scalable, and accurate solution, with help from other technologies, such as cloud computing, machine learning, and so on. In other words, 5G is key in smart cities because this generation optimizes how to transmit and receive data and performs the way to obtain data. Based on these data, electronic devices can make accurate decisions after processing.

Table 8 was divided into parts, in which the first part deals exclusively with the description and impact on 5G, and the second part describes cases and ways to prevent this interference.

It is pertinent to consider how this type of interference can affect or impact 5G networks today, since it must be kept in mind with the explosive development of technology. In turn, this can affect different 5G architectures, since the main objectives of these networks are to improve things such as latency, throughput, transmission speeds, etc.

Table 9 provides an overview of different types of interference that can occur in 5G networks and their potential impacts on network performance. Adjacent Channel Interference occurs when the frequency channel of our device is overlapped with nearby frequencies, leading to interference. Intra-cell and inter-cell interference occurs when multiple users in neighboring cells attempt to use the same frequency simultaneously or when there is a simultaneous short-range transmission by two base stations in the same cell. This type of interference can severely damage 5G networks, particularly in architectures where there are micro- or macro-cells. Intra-channel and inter-channel interference occurs when there is little power within the macro network cell, though inter-channel interference can also occur due to the proximity of devices operating at different frequencies. Inter-Symbol interference occurs when one or more symbols interfere with other symbols, resulting in signal distortion due to phase or amplitude dispersion of the channel. Inter-Carrier interference occurs when the signal is lost due to an offset between the subcarriers, leading to large-scale or small-scale fading and variations in the received signal. Inter-Numerology interference occurs when using the numerology system for greater flexibility, leading to misalignments or imperfections known as inter-numerology interference. Cross-Link Interference occurs when a transmission is made to a neighboring cell in different directions simultaneously, leading to performance failures. Inter-Beam interference occurs when using MIMO technology, with the spatial division of the multiple beams causing interference from adjacent beams either by the same cell or by neighboring cells. Finally, Multi-User interference occurs when multiple users try to transmit information simultaneously, leading to problems during user management in uncoordinated cells.

## 4. Results and Discussions

In recent years, the deployment of 5G networks has gained significant attention due to its potential to revolutionize the communication industry. One of the areas where 5G networks are expected to have a substantial impact is the IoT services. This literature review article aims to provide an in-depth analysis of the impact of 5G networks on IoT services, specifically examining the issue of interference in this type of network and its related technologies.

As a result of new technologies in mobile communications, 5G can provide a solution in a smart city context. Each IoT device can consume a reduced bandwidth, but the problem arises when the number of IoT devices increases. For this reason, it is essential to provide high beam width for better communication between IoT devices.

The emergence of 5G networks and Internet of Things services has brought about a new era of connectivity and transformation to various industries. With the increasing number of connected devices, there is a need for a network that can handle the massive data transfer, low latency, and high-speed communication required by IoT devices. The convergence of 5G networks and IoT services is expected to revolutionize the way devices communicate with each other and the internet. In this section, we will critically discuss the related impact of network and service convergence between 5G networks and IoT services.

One of the main benefits of the convergence of 5G networks and IoT services is the ability to support massive machine-to-machine communication. The integration of 5G networks with IoT devices creates a seamless connection, allowing devices to communicate with each other and the internet at high speeds and low latency. This has significant implications for industries such as healthcare, transportation, and manufacturing, where large volumes of data need to be transmitted in real time to enable efficient operations. For instance, connected cars, trains, and airplanes can communicate with each other and other connected devices in real time, leading to improved safety and efficiency.

Another impact of the convergence of 5G networks and IoT services is the creation of new business models and revenue streams. With the increased speed and capacity of 5G networks, service providers can offer new IoT services such as smart homes, smart cities, and smart factories. This creates an opportunity for service providers to develop new business models and revenue streams, such as selling data insights, providing managed services, and offering customized solutions. For example, telecom operators can offer IoT connectivity as a service, which provides businesses with a cost-effective and scalable method to connect and manage their IoT devices.

However, the convergence of 5G networks and IoT services also presents some challenges that need to be addressed. One of the challenges is the issue of security and privacy. As the number of connected devices grows, the potential for cyberattacks and data breaches also increases. Therefore, service providers and device manufacturers need to work together to ensure that IoT devices are secure and comply with data protection regulations. Additionally, the convergence of 5G networks and IoT services requires significant investments in infrastructure and technology. Service providers need to deploy a massive number of 5G base stations to enable reliable and consistent connectivity for IoT devices.

The convergence of 5G networks and IoT services has the potential to revolutionize the way devices communicate with each other and the internet. It has significant implications for industries such as healthcare, transportation, and manufacturing, creating new business models and revenue streams. However, it also presents challenges such as security and privacy concerns and the need for significant investments in infrastructure and technology. Therefore, service providers and device manufacturers need to work together to address these challenges and ensure that the convergence of 5G networks and IoT services leads to a safer, more efficient, and connected world.

Table 10 discusses the potential positive and negative impacts of integrating 5G technologies with IoT solutions. The table is divided into three columns, including Category, Context, and Explanation.

The first category mentioned in the table is Positive, which includes three different contexts. The first context is Cost Optimization, which refers to the potential cost savings associated with integrating 5G technologies with IoT solutions. With the help of data gathered by devices, businesses and organizations can optimize their processes, leading to more cost-effective solutions. Additionally, data-driven decision making allows private or public entities to make better decisions. For instance, a city can use IoT solutions to optimize traffic flow, leading to reduced fuel consumption and fewer traffic jams.

The second context mentioned in the Positive category is Improving QoS, which refers to the potential improvement in the quality of life for citizens. By providing relevant information and real-time control over public infrastructure, cities can positively impact citizens’ lives. For example, using IoT solutions, a city can improve security by monitoring public spaces and providing real-time alerts in case of emergencies. Additionally, cities can create new economic development opportunities fueled by the digitalization era.

The third context mentioned in the Positive category is Reducing climate change. By implementing IoT solutions with 5G technologies, businesses can have closer control over industrial processes and real-time analytics regarding relevant environmental properties. This can be the next step in becoming a more environmentally friendly society.

Moving onto the Negative category, the table mentions two different contexts. The first context is Digital Non-Inclusion, which refers to the potential negative impact of IoT solutions on regions with less access to technological services, while 5G and IoT efforts can improve the quality of life of those cities that can afford it, they currently do not offer cost-effective solutions for regions with less access to technological services.

The second context mentioned in the Negative category is privacy compromises. Having real-time information on assets, people, and any living or non-living organism in a region can seriously threaten privacy violations. Political or social movements can negatively use sensible data related to citizens to perform any action.

Figure 10 discusses the main challenges on 5G networks. One of the main challenges of 5G networks related to interference, IoT devices, and network optimization is the high demand for wireless connectivity and data transmission. With the increasing number of IoT devices and the exponential growth of data usage, 5G networks must be optimized to handle the massive amount of data transmission while minimizing interference. The deployment of small cells and heterogeneous networks is one strategy to increase network capacity and reduce interference. However, the complexity of network planning, deployment, and management increases with the deployment of small cells, which requires sophisticated optimization techniques. Additionally, the development of 5G IoT devices with limited power and processing capabilities also presents a challenge. The design of low-power wireless communication protocols and efficient resource allocation techniques is necessary to optimize the performance of IoT devices in 5G networks. 5G networks’ challenges related to interference, IoT devices, and network optimization require advanced solutions and innovative technologies to ensure high-quality wireless connectivity and meet the demands of the growing digital ecosystem.

Table 11 is a representation of the future research challenges in 5G that are directly related to IoT services. It is a five-column table with the following labels for each column: Features, Advantages, Research Challenges, Key Requirements, and Interoperability. The Features column lists the main features that are relevant for IoT services in 5G. In this table, the features are IoT Services, Edge Computing, 5G Radio Access, and Network Slicing. The Advantages column lists the advantages that 5G can provide to IoT services, for example, enabling new use cases, reducing latency, providing high data rates, and providing customizable networks. The Research Challenges column lists the key research challenges that need to be addressed in order to fully realize the potential of 5G for IoT services, for example, network slicing, security, scalability, resource allocation, and spectrum management. The Key Requirements column lists the key technical requirements that need to be met in order to address the research challenges, for example, low latency, high reliability, energy efficiency, and service level agreements. The Interoperability column lists the key interoperability issues that need to be addressed in order to ensure that different 5G networks and IoT devices can work together seamlessly, for example, standardization, integration with existing systems, compatibility with different technologies, and interoperability between network slices.

Interference can have a significant impact on 5G networks, particularly in the context of Internet of Things (IoT) devices. The large number of IoT devices that will be connected to 5G networks will inevitably lead to increased interference, which can result in degraded network performance and poor user experience.

One of the critical responses to interference in 5G networks is to employ advanced signal processing techniques to mitigate the effects of interference. For example, beamforming and advanced receiver algorithms can help to reduce interference on 5G networks. Additionally, the use of multi-antenna systems and smart antennas can improve the quality of the received signal and reduce interference.

Another critical response is to implement interference management protocols that enable the efficient use of available radio resources. These protocols can help to coordinate the use of different channels, reduce interference between neighboring cells, and optimize the use of spectrum resources.

For instance, the use of small cell networks can help to reduce interference by reducing the distance between the transmitter and receiver. In addition, the use of frequency reuse techniques, such as fractional frequency reuse, can help to mitigate the impact of interference in multi-cell networks.

Table 12 summarizes the future research challenges in 5G for IoT services. The first item, Issues, describes the main challenges faced in the implementation of 5G for IoT services. The five issues identified in the table are Interference, Security, Energy Efficiency, Scalability, and Latency. The item Methodologies presents the proposed techniques or methods to address the identified issues. For example, DSA and Cooperative Sensing are proposed to address the Interference issue. Similarly, Authentication and Encryption are proposed to address the Security issue. The table provides a list of techniques for each of the identified issues. The item Advantages describes the potential benefits of using the proposed methodologies. For instance, the use of DSA and Cooperative Sensing can lead to better spectrum utilization and increased reliability. Similarly, using authentication and encryption can provide secure communication and protect against attacks. The item Limitations/Future Work describes the challenges or limitations of the proposed methodologies and highlights the areas that need further research. For example, the table highlights the need for developing efficient and scalable DSA and cooperative sensing algorithms to address the Interference issue. Similarly, developing lightweight and energy-efficient security mechanisms is identified as future work to address the Security issue.

Table 13 includes several types of interference that affect 5G networks and IoT services. These include atmospheric absorption, free space path loss, reflection, refraction, diffraction, scattering, rain fade, multipath fading, co-channel interference, adjacent channel interference, and interference from other radios and IoT devices. The table provides technical details about the frequency range, bandwidth, power level, and impact of each interference type.

Interference: The type of interference that affects 5G networks and IoT services.Frequency: The frequency range in which the interference occurs.Bandwidth: The bandwidth of the interference.Power: The power level of the interference.Impact: The effect of the interference on 5G networks and IoT services.

## 5. Conclusions

The convergence of 5G technology and the Internet of Things is an essential step towards achieving new business opportunities and improving connectivity worldwide. The ecosystem of IoT devices is complex, and choosing the right primary or complementary connectivity option depends on factors such as deployment costs, range, interference, and capabilities. Technical studies have shown that 5G and other services can coexist in specific frequency bands, provided that the technical conditions are adequately adapted. Compliance with the permitted exposure limits is also essential when designing new applications or technological accessories using 5G and IoT.

The convergence of networks and services, driven by 5G technology, is transforming the internet into a complex and multifaceted ecosystem integrated into nearly every aspect of our daily lives. The availability and speed of access to the internet are increasing, allowing users to access high-quality media content in real-time and internet service providers to offer a range of new services and applications. Cellular connectivity will enable the achievement of key IoT goals, such as reducing device complexity and cost, increasing coverage to support remote applications, and providing deployment flexibility, high capacity, and long battery life.

Businesses can benefit greatly from the optimization of network performance in 5G networks, which is essential for adequately functioning business processes, improving productivity, reducing downtime, and enhancing customer satisfaction. The potential of the convergence of networks and services in increasing the availability and speed of access to the internet enables a range of new and innovative applications and services, transforming the way we live and work.

Managing interference in 5G networks is a significant challenge in ensuring the reliability and performance of IoT services. Effective interference management techniques, diverse connectivity and latency requirements of IoT devices and applications, and external interference are significant issues that must be addressed to ensure that 5G networks can support the massive number of devices and applications that rely on them.

5G has a lot of features necessary for smart cities. However, it can be improved by combining it with protocols such as LoRaWAN, which provides low-range communication, Z-Wave, Zigbee, and SigFox for house IoT devices. 5G with IoT technologies combine all this to provide a more complex solution, taking care of the attenuation of the signal, security protocols, bandwidth, QoS, and more.

## Figures and Tables

**Figure 1 sensors-23-03876-f001:**
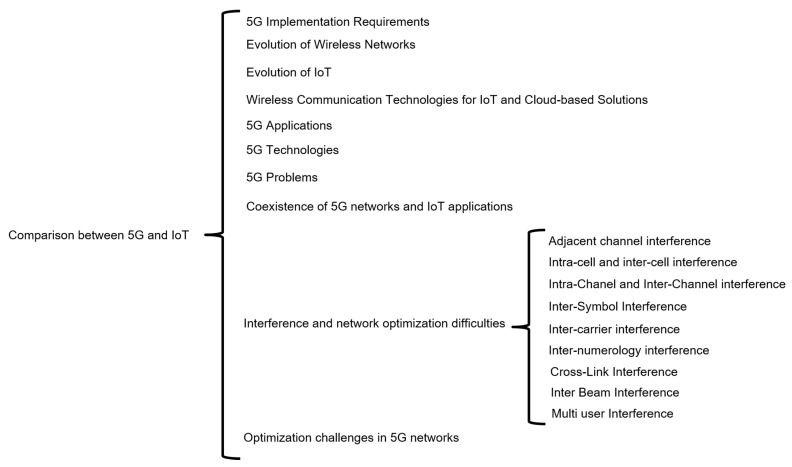
Summary of conceptual schemes that addresses the present work.

**Figure 2 sensors-23-03876-f002:**
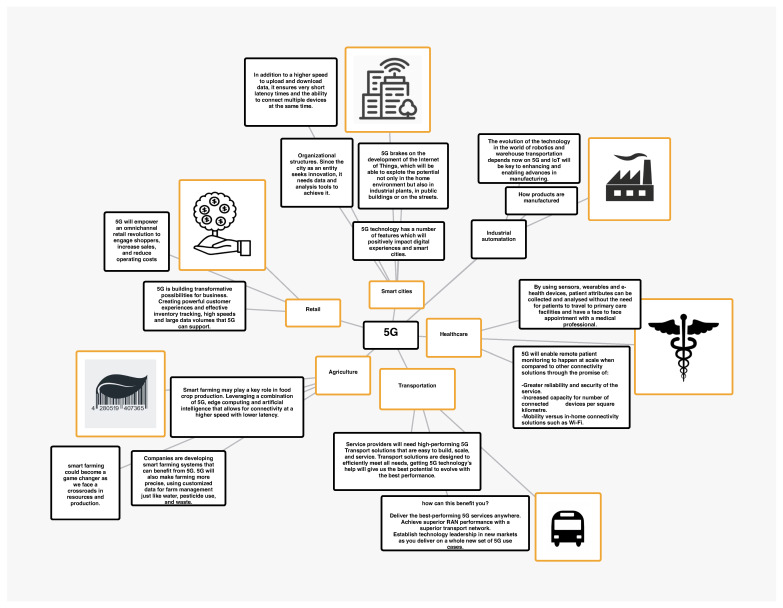
5G applications impacted by IoT devices.

**Figure 3 sensors-23-03876-f003:**
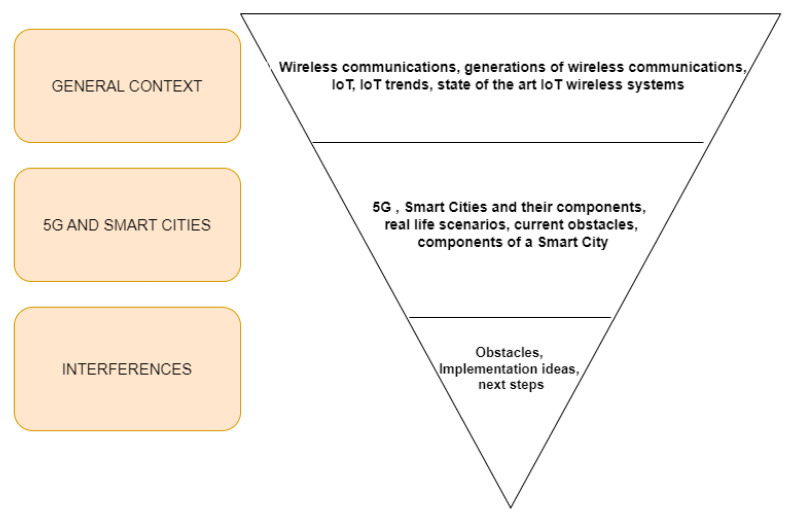
5G and IoT technologies’ impact scheme.

**Figure 4 sensors-23-03876-f004:**
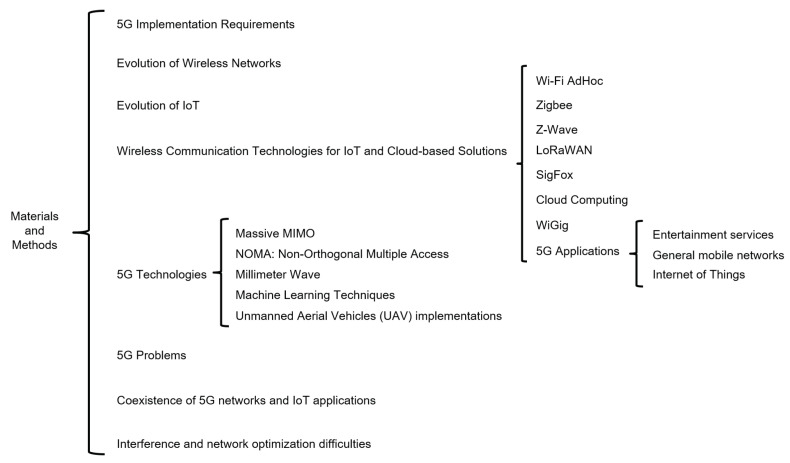
Materials and Methods for 5G and IoT services.

**Figure 5 sensors-23-03876-f005:**
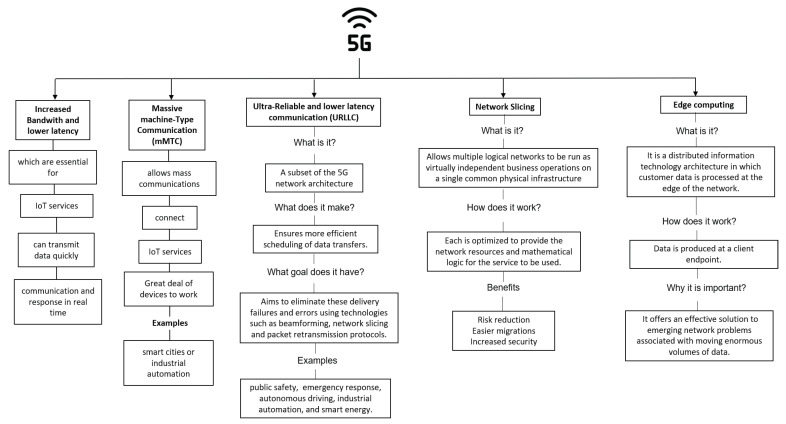
5G features that are essential for the Internet of Things services.

**Figure 6 sensors-23-03876-f006:**
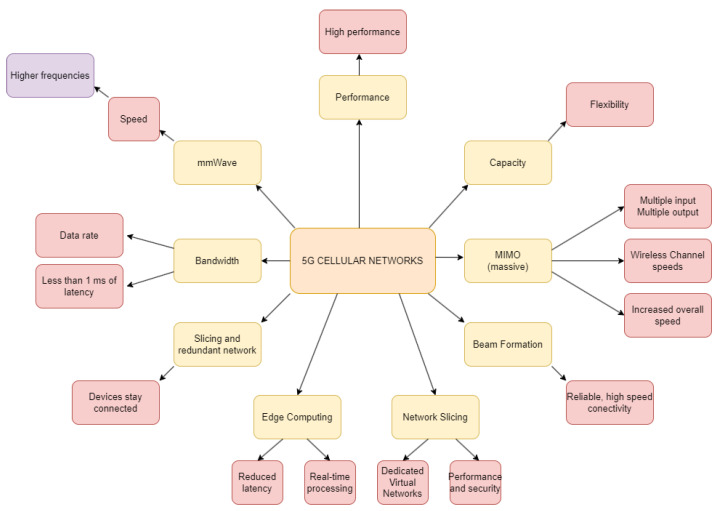
5G implementation requirements.

**Figure 7 sensors-23-03876-f007:**
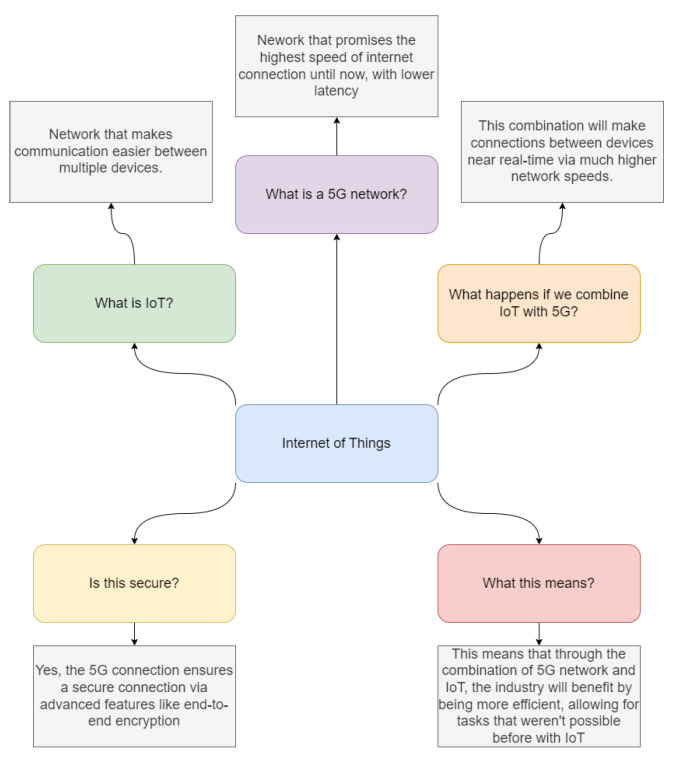
Defining 5G and features that are essential for the Internet of Things services.

**Figure 8 sensors-23-03876-f008:**
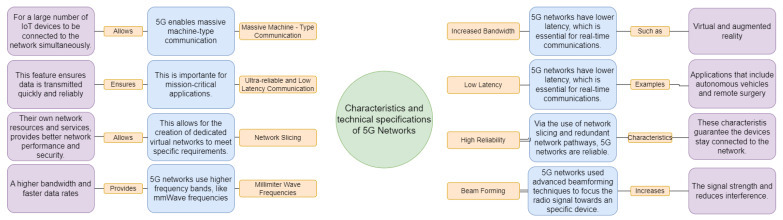
Characteristics and technical specifications of 5G networks.

**Figure 9 sensors-23-03876-f009:**
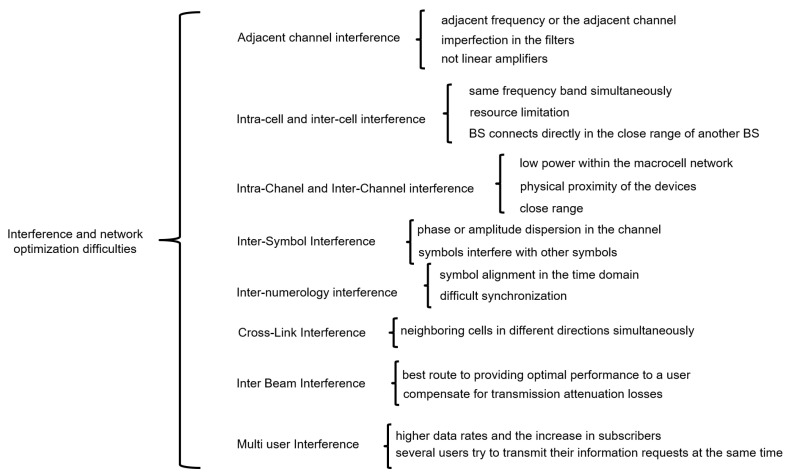
Main concepts of interference in 5G.

**Figure 10 sensors-23-03876-f010:**
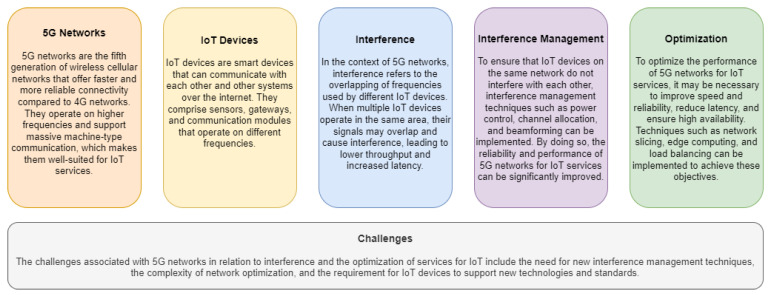
Challenges of 5G networks.

**Table 2 sensors-23-03876-t002:** Comparison of problems presented by 5G and IoT services.

Problem	5G	IoT Services
Security	5G networks are vulnerable to various security threats, such as DDoS attacks, identity theft, and man-in-the-middle attacks.	IoT devices are susceptible to security breaches due to poor encryption, weak passwords, and outdated firmware.
Latency	5G networks have lower latency, which can be a problem for certain IoT applications that require real-time response.	IoT services may suffer from latency due to network congestion, distance from the server, and the number of devices connected to the network.
Interference	5G networks can experience interference from other wireless devices, which can disrupt the transmission of data.	IoT services may also suffer from interference due to environmental factors, such as obstacles and interference from other wireless devices.
Cost	The cost of 5G infrastructure and services may be prohibitively high for many IoT applications, particularly those that require large-scale deployment.	IoT services may also be expensive to deploy and maintain, particularly if they require high bandwidth or specialized hardware.
Compatibility	Some IoT devices may not be compatible with 5G networks, which can limit their usefulness in certain applications.	IoT services may also be limited by compatibility issues, particularly if they rely on proprietary hardware or software.

**Table 3 sensors-23-03876-t003:** Coexistence of 5G services and networks and Internet of things applications.

Service	Wireless Technology	Algorithms Used	Coexistence of Services	Applications and System Description
5G NR [70]	NR-U	NOMA	Coexisting with 4G and Wi-Fi	Ultra-reliable and low-latency communications for industrial automation and autonomous vehicles
5G-V2X [71]	PC5	OFDMA	Coexisting with 4G and Wi-Fi	Vehicle-to-everything communications for traffic management and safety
5G mMTC [72]	NR-MTC	NOMA	Coexisting with 4G and Wi-Fi	Massive machine-type communications for smart cities and agriculture
5G-UHD [73]	NR-U	OFDMA	Coexisting with 4G and Wi-Fi	Ultra-high-definition video communications and virtual reality
5G-IoT [74]	NR-MTC	NOMA	Coexisting with 4G and Wi-Fi	Internet of things communications for smart homes and wearables
5G-eMBB [75]	NR-eMBB	OFDMA	Coexisting with 4G and Wi-Fi	Enhanced mobile broadband for high-speed data transfer and streaming
5G-URLLC [76]	NR-URLLC	NOMA	Coexisting with 4G and Wi-Fi	Ultra-reliable and low-latency communications for mission-critical applications
5G-mIoT [77]	NR-mIoT	NOMA	Coexisting with 4G and Wi-Fi	Massive IoT communications for smart cities and agriculture
5G-gNB [78]	NR-gNB	OFDMA	Coexisting with 4G and Wi-Fi	Gigabit-class communications for high-speed data transfer and streaming
5G-Slicing [79]	NR-Slicing	NOMA	Coexisting with 4G and Wi-Fi	Network slicing for multiple services and applications

**Table 4 sensors-23-03876-t004:** Types of interference that impact 5G networks and IoT.

Interference Type	Coverage	Latency	Availability	Access	Modulation	Coding
Noise	Reduced	Increased	Reduced	Reduced	Reduced	Reduced
Multipath	Reduced	Increased	Reduced	Reduced	Reduced	Reduced
Inter-cell Interference	Reduced	Increased	Reduced	Reduced	Reduced	Reduced
Co-channel Interference	Reduced	Increased	Reduced	Reduced	Reduced	Reduced
Interference from other devices	Reduced	Increased	Reduced	Reduced	Reduced	Reduced
Jamming	Reduced	Increased	Reduced	Reduced	Reduced	Reduced
Interference from other wireless networks	Reduced	Increased	Reduced	Reduced	Reduced	Reduced

**Table 5 sensors-23-03876-t005:** Mobile generations and IoT.

Year	Technology	Impact in IoT
2000	3G allows devices to connect to the internet since 3G enables mobile and wireless internet connections.	With internet connections, electronic devices start transmitting data through the internet, the first step to developing IoT devices.
2008	4G enables cloud computing technology and transmits information with the IP protocol. In addition, 4G increased the bandwidth of each transmission.	The data transmission through IP protocol enables an easy communication method with electronic devices. The cloud enables a way to develop more affordable solutions. Furthermore, the internet connection cost decreases because 4G delivers a cheap way to transmit data as IP protocols manage the data more efficiently. This generation is the most important in an IoT context because the industry has all the resources to connect devices to the cloud.
2019	With 5G, an essential feature is the beam width, an important key in the IoT context.	If we have more bandwidth, we can provide better solutions, such as real-time monitoring systems. In our society, these solutions are vital if we want to automate processes. For this reason, 5G plays an essential role in the IoT context because the industry is trying to automate all its processes.

**Table 6 sensors-23-03876-t006:** Important Features in 3G, 4G, and 5G.

	Generation	3G	4G	5G
Feature				
Standard		WCDMA, CDMA2000	OFDMA, MC-CDMA	CDMA, BDMA
Data rate		2 Mbps	2 Mbps–1 Gbps	1 Gbps and higher
Frequency		1.8–2.5 GHz	2–8 GHz	3–300 GHz
Core type network		Packet network	All IP network	IP network and 5G-NI

**Table 7 sensors-23-03876-t007:** Expectation for 5G in IoT.

Feature	Description
Bandwidth	More bandwidth to supply applications with high speed.
AI	Artificial intelligence can analyze all the data generated by the IoT devices to take accurate decisions.
Real-time monitoring and management	The capability to monitor and manage electronic devices remotely.
Swarm intelligence	Each IoT device can be a node, and working with other nodes, can work as a swarm intelligence.
QoS	5G can provide a better QoS in IoT connections.
Low Latency	5G reduces the latency, which means that the IoT devices can take immediate decisions.
Cloud computing	5G can provide better connections to the cloud, which means each IoT device can use all the resources of the cloud.

**Table 8 sensors-23-03876-t008:** Technologies for the elderly.

Interference	Description	5G Impact
Adjacent Channel Interference	This occurs when the frequency channel of our device is overlapped.	If we do not have good bandpass filters, there will be interference from nearby frequencies. In 5G devices, this can have a significant impact.
Intra-cell and inter-cell interference	Inter-cell interference occurs when two users in neighboring cells attempt to use the same frequency simultaneously. Intra-cell interference occurs when there is a simultaneous short-range transmission by two BSs, with distortion from the user and the other equipment in the same cell.	This type of interference can severely damage 5G networks because for bases or architectures where there are micro- or macro-cells, the power of neighboring signals in both uplink and downlink transmission by multiple users can interfere with each other [98].
Intra-Channel and Inter-Channel interference	Intra-channel interference occurs when there is little power within the macro network cell.	Inter-channel interference occurs due to the proximity of devices when two separate frequencies cause interference. As they operate at short range, the transmitter of a high-power signal causes interference; interference can also be due to the exploration of hetnets with OFDM. There are ICI reduction solutions with reverse frequency allocation (RFA) employed, which is a proactive interference.
Inter-Symbol interference	This type of interference occurs when one or more symbols interfere with other symbols.	It is caused by phase or amplitude dispersion of the channel, which results in signal distortion. This can be clearly seen in OFDMA, which causes multipath propagation and impacts bandwidth efficiency.
Inter-Numerology interference	Occurs when using the numerology system for greater flexibility and does not allow the alignment in the time domain to be perfect. These misalignments or imperfections are known as inter-numerology interference.	In the search for higher performance and significant bandwidth, using numerology, can cause interference by having networks with many devices sending significant amounts of information, which makes this type of interference more pronounced, affecting the network’s performance.
Cross-Link Interference	This type of interference occurs when a transmission is made to a neighboring cell in different directions simultaneously, either by arbitrarily overlapping resources or by time frequency.	Cross-link interference can affect 5G networks due to the amount of BS required, since incorrect hopping can cause performance failures.
Inter-Beam interference [99]	This occurs when using MIMO technology, as this type of array sends RF energy in all directions, with the spatial division of the multiple beams causing interference. Adjacent beams cause this either by the same cell or by neighboring cells.	It causes interference in applications where MIMO antenna technology is introduced due to the same usage. When looking for better spectral efficiency and improvements in network performance, interference from adjacent beams within the same cell or multiple MIMO arrays may negatively influence their neighbors.
Multi-User interference [100]	This occurs when, in MU-MIMO, multiple users try to transmit information simultaneously.	Considering that in 5G, performance and information forwarding improvements are sought, when switching to MU-MIMO, uncoordinated cells encounter more significant problems during user management.
Adjacent Channel Interference	This occurs when the frequency channel of the device is overlapped.	If we do not have good bandpass filters, there will be interference from nearby frequencies. In 5G devices, this can have a significant impact.

**Table 9 sensors-23-03876-t009:** Interference and its impact on 5G networks.

Interference Type	Description	Impact on 5G Networks (✔ yes, × is no
Adjacent Channel	Frequency channel overlap	✔
Intra-cell and inter-cell	Simultaneous transmission in neighboring cells	×
Intra-Channel and inter-Channel	Low power in the macro network cell	✔
Inter-Symbol	Interference between symbols	×
Inter-Carrier	Signal loss due to subcarrier offset	×
Inter-Numerology	Misalignments in the numerology system	×
Cross-Link	Overlapping transmissions to neighboring cells	×
Inter-Beam	Interference from adjacent beams in MIMO	×
Multi-User	Simultaneous transmission from multiple users	×

**Table 10 sensors-23-03876-t010:** Explanation of the positive or negative impact in 5G and IoT.

Category	Context	Explanation
Positive	Cost Optimization	Integrating 5G technologies with IoT solutions will allow for more cost-effective solutions when 5G technologies reach the mainstream market. Optimizing processes with the data gathered by the devices will allow for data-driven decision making, allowing private or public entities to make better decisions.
Positive	Improving QoS	By providing relevant information and real-time control over public infrastructure, citizens will be positively impacted by reduced traffic, improved security, and new economic development opportunities fueled by the digitalization era.
Positive	Reducing climate change	Having closer control over industrial processes and real-time analytics regarding relevant environmental properties, implementing IoT solutions with 5G technologies can be the next step in becoming a more environmentally friendly society.
Negative	Digital Non-Inclusion	While 5G and IoT efforts can improve the quality of life of those cities that can afford it, they currently do not offer cost-effective solutions for regions with less access to technological services.
Negative	Privacy compromises	Having real-time information on assets, people, and any living or non-living organism in a region can seriously threaten privacy violations. Political or social movements can negatively use sensible data related to citizens to perform any action.

**Table 11 sensors-23-03876-t011:** Future research challenges in 5G related to IoT services.

Features	Advantages	Research Challenges	Key Requirements	Interoperability
IoT Services	Enables new use cases	Network slicing	Low latency	Standardization
		Security	High reliability	Integration with existing systems
		Scalability	Massive machine-type communication	Compatibility with different technologies
Edge Computing	Reduced latency	Resource allocation	Energy efficiency	Interoperability with cloud services
		Edge intelligence	Resource management	Security
		Edge analytics	QoS management	Integration with network slicing
5G Radio Access	High data rates	Spectrum management	Low power consumption	Compatibility with legacy systems
		Multi-connectivity	Interference management	Network densification
		mmWave communications	Coverage	Integration with edge computing
Network Slicing	Customizable networks	Orchestration	Service level agreements	Interoperability between slices
		Resource allocation	Isolation	Scalability
		QoS management	Security	Integration with existing networks

**Table 12 sensors-23-03876-t012:** Future research challenges in 5G for IoT services.

Issues	Methodologies	Advantages	Limitations/Future Work
Interference	Dynamic Spectrum Access, cooperative Sensing	Better spectrum utilization, increased reliability	Developing efficient and scalable DSA (Dynamic Spectrum Access) and cooperative sensing algorithms
Security	Authentication, encryption	Secure communication, protecting against attacks	Developing lightweight and energy-efficient security mechanisms
Energy Efficiency	Power management, resource allocation	Longer battery life, improved system capacity	Developing energy-efficient algorithms for resource allocation and power management
Scalability	Network slicing, virtualization	Better resource utilization, improved service quality	Developing efficient network slicing and virtualization techniques for massive IoT deployments
Latency	Edge computing, network architecture	Reduced communication delay, improved application performance	Developing low-latency edge computing and network architecture for IoT services

**Table 13 sensors-23-03876-t013:** Interference characteristics in 5G networks and IoT services.

Interference	Frequency	Bandwidth	Power	Impact	Reference
Atmospheric Absorption	24–40 GHz	Narrowband	Low	Attenuation	[101]
Free Space Path Loss	All	All	Low	Attenuation	[102]
Reflection	All	All	Low	Multipath Fading	[103]
Refraction	All	All	Low	Path Bending	[104]
Diffraction	All	All	Low	Path Bending	[105]
Scattering	All	All	Low	Multipath Fading	[106]
Rain Fade	10–100 GHz	Wideband	High	Attenuation	[107]
Multipath Fading	All	All	Low	Intersymbol Interference	[108]
Co-Channel Interference	All	All	High	Reduced Signal Quality	[109]
Adjacent Channel Interference	All	All	High	Reduced Signal Quality	[110]
Interference from Other Radios	All	All	High	Reduced Signal Quality	[111]
Interference from Other IoT Devices	All	All	Low	Reduced Signal Quality	[112]

## Data Availability

Not applicable.

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
