# Peer review of "Utilization of 5G Technologies in IoT Applications: Current Limitations by Interference and Network Optimization Difficulties—A Review"

_sensors, 2023, doi:10.3390/s23083876_

Round 1

Reviewer 1 Report

Please see the attached comments.

Author Response

Dear

Editor

Sensors

We are submitting the paper:

“Utilization of 5G Technologies in IoT Applications: Current limitations by interference and network optimization difficulties - A Review”

Authored by: Mario Pons1, Estuardo Valenzuela1, Brandon Rodríguez1, Juan Arturo Nolazco-Flores2, and Carolina Del-Valle-Soto3,*

We would like to thank the reviewers and editors for their detailed analysis of the manuscript; the comments are very valuable to us. In the revised version of the paper, we have incorporated the all changes recommended by the reviewers.

Comments to all observations and suggestions including point-by-point responses are addressed in the following text.

Reviewer 1 comments

Comment 1: This paper presented the 5G and IoT technologies, explaining common architectures, typical IoT implementations, and recurring problems. The authors also presented a detailed and explained overview of interference in general wireless applications, interference unique to 5G and IoT, and possible optimization techniques to overcome these challenges. Generally speaking, the authors have done a solid survey on 5G technologies in IoT applications. However, some parts need to be further clarified. The reviewer’s concerns are listed as follows.

  1. Honestly, the whole paper is well written and the authors provided a full overview of interference limitation for 5G techniques in IoT application.

Response: Many thanks to the Reviewer for his/her invaluable interest in the comments on this manuscript.

Comment 2: As 36-page manuscript, the content is a little bit long, it is suggested to add a system framework diagram of the whole paper to make it more clear for readers.

Response: Many thanks to the Reviewer. The Reviewer is right, and we have included a Figure that frames the entire summary of the manuscript, where the relationships between the main concepts are explained in an orderly manner.

Figure 1 shows the comparison between 5G and IoT, that is a topic of interest in the technology world. To implement 5G, there are specific requirements, including infrastructure and specialized hardware, while the evolution of wireless networks has led to faster and more reliable data transfer rates. The evolution of IoT has enabled the creation of low-power, low-cost devices that can be connected to the internet. Wireless communication technologies like Bluetooth, Wi-Fi, ZigBee, and LoRa are suitable for different types of IoT applications, and cloud-based solutions are used to store and process vast amounts of data generated by IoT devices. 5G applications include autonomous vehicles, remote surgery, and virtual and augmented reality, and these require high bandwidth, low latency, and reliable connectivity. However, there are challenges associated with 5G networks, such as interference and network optimization difficulties. The coexistence of 5G networks and IoT applications is a concern, and optimization challenges in 5G networks need to be addressed to enable efficient and effective implementation.

Comment 3: The font of some figures, such as figure 1, 3 and 6, are too small to figure out. The authors can change the architecture of figures to increase the font.

Response: Thanks to Reviewer. We have improved the font of the Figures you mention to make the paper more readable.

Comment 4: The contributions are not well introduced, the authors need to improve the contributions to show the advantages and differences of this survey compared to the similar surveys.

Response: The Reviewer's point is very coherent; therefore, we have added a specific paragraph in the Motivation, highlighting the contribution of the work.

The main contribution of this review is to present the idea that the convergence of 5G networks and IoT services represents a technological revolution that promises to change how we interact with the world around us. However, this union has its challenges, and only by overcoming them can we unlock the full potential of these groundbreaking technologies. By bridging the gap between 5G and IoT, we pave the way for a new era of innovation: every device is connected, and every experience is seamless. This work describes the main challenges between 5G networks and IoT services and highlights the need for seamless integration of these technologies to achieve their full potential. This paper reinforces the idea that IoT technologies power 5G services, and their compatibility is crucial for offering high-speed and low-latency services to consumers. Furthermore, it underscores that interference is one of the biggest problems facing manufacturers, and they must consider it to offer compatible and reliable services. Ultimately, this article guides the industry to ensure that 5G and IoT technologies work together seamlessly, opening up endless possibilities for innovation and progress.

Comment 5: It is suggested to introduce the following recent works in 5G technologies, such as RIS [R1] and massive MIMO [R2], IoT [R3] and NOMA [R4] fields to highlight the state-of-the-art of this paper.

[R1] “Refracting RIS aided hybrid satellite-terrestrial relay networks: Joint beamforming design and optimization,” IEEE Transactions on Aerospace and Electronic Systems, vol. 58, no. 4, pp. 3717-3724, Aug. 2022.

[R2] “Secrecy-energy efficient hybrid beamforming for satellite-terrestrial integrated networks,” IEEE Transactions on Communications, vol. 69, no. 9, pp. 6345-6360, Sep. 2021.

[R3] “Supporting IoT with rate-splitting multiple access in satellite and aerial-integrated networks,” IEEE Internet of Things Journal, vol. 8, no. 14, pp. 11123-11134, Jul. 2021.

[R4] “Joint beamforming and power allocation for satellite-terrestrial integrated networks with non-orthogonal multiple access,” IEEE Journal of Selected Topics in Signal Processing, vol. 13, no. 3, pp. 657-670, June 2019

Response: Many thanks to the Reviewer because the references are very accurate and relevant. We have added all the references you recommended and highlighted them in the section so you can quickly identify them.

Comment 6: The authors should add some principles diagram of some key techniques or add some simulations to enhance the readability.

Response: Thanks to the Reviewer for his wise comment. We have added these figures to make the manuscript interactive and show interesting conceptual relationships.

Figure 4 summarizes this section, where the main concepts related to the impact of 5G networks on IoT services are observed. We analyze the implementation, the evolution, the types of wireless technologies that influence current services, the leading 5G technologies and problems, and the coexistence of next-generation networks. We frame interference as the heart of the study of problems in this type of network.

Figure 9 covers various types of interference in wireless communication systems in 5G. Adjacent channel interference occurs when adjacent frequencies or channels overlap due to imperfections in filters or non-linearity in amplifiers. Inter-cell interference happens when two users from neighboring cells try to use the same frequency band simultaneously due to resource limitation within the network. Intra-channel and inter-channel interference occur due to physical proximity of devices and low power within the macrocell network, respectively. Inter-symbol interference results from signal distortion due to phase or amplitude dispersion in the channel. Inter-carrier interference occurs due to frequency offsets between sub-carriers. Inter-numerology interference arises due to non-orthogonality in the system, causing difficulties for symbol alignment. Cross-link interference occurs when signals are transmitted to neighboring cells in different directions simultaneously. Inter-beam interference happens due to the use of multi-beam antenna systems, causing interference from adjacent beams. Multi-user interference happens when several users try to transmit their information requests at the same time, and MU-MIMO is used to increase the capacity and performance of wireless broadcasting systems.

Thank you very much.

Sincerely,

Carolina Del-Valle-Soto

Universidad Panamericana. Facultad de Ingeniería. Álvaro del Portillo 49, Zapopan, Jalisco, 45010, México.

Phone: +52 (33) 13682200 | Ext. 4866

Reviewer 2 Report

It is a rather long, well-organized review work, bringing together the main coexistence technologies and interference mitigation for 5G. I consider the work a valuable scientific contribution to this area of ​​knowledge. My only suggestion is about the limited approach over the millimeter wave range, in particular the absence of technologies such as WiGig (802.11ay). The Release 17 of 3GPP worked on NR operation up to 71GHz.

Author Response

Dear

Editor

Sensors

We are submitting the paper:

“Utilization of 5G Technologies in IoT Applications: Current limitations by interference and network optimization difficulties - A Review”

Authored by: Mario Pons1, Estuardo Valenzuela1, Brandon Rodríguez1, Juan Arturo Nolazco-Flores2, and Carolina Del-Valle-Soto3,*

We would like to thank the reviewers and editors for their detailed analysis of the manuscript; the comments are very valuable to us. In the revised version of the paper, we have incorporated the all changes recommended by the reviewers.

Comments to all observations and suggestions including point-by-point responses are addressed in the following text.

Reviewer 2 comments

Comment 1: It is a rather long, well-organized review work, bringing together the main coexistence technologies and interference mitigation for 5G. I consider the work a valuable scientific contribution to this area of ​​knowledge. My only suggestion is about the limited approach over the millimeter wave range, in particular the absence of technologies such as WiGig (802.11ay). The Release 17 of 3GPP worked on NR operation up to 71GHz.

Response: Many thanks to the Reviewer for his/her invaluable comments. You are right, we have added a subsection, and in the conceptual figures, we have also added a schematic dedicated to WiGig. Fascinating this missing concept of the Reviewer.

WiGig, also known as 802.11ay, is a wireless communication technology that operates on the 60 GHz frequency band [52]. It was developed as an extension of the Wi-Fi standard to provide high-speed, short-range wireless communication, primarily for applications that require high bandwidth, such as virtual reality, high-definition video streaming, and gaming. WiGig supports multi-gigabit data transfer rates, with theoretical speeds of up to 176 Gbps, which is much faster than the previous Wi-Fi standards [53]. It achieves this speed through the use of wider bandwidth and advanced modulation techniques such as Quadrature Amplitude Modulation (QAM) and Orthogonal Frequency Division Multiplexing (OFDM). Another notable feature of WiGig is its low-latency, making it ideal for applications that require real-time data transfer, such as gaming and virtual reality [54]. It also supports multiple-input multiple-output technology, which enables multiple antennas to transmit and receive data simultaneously, improving the overall performance and efficiency of the network. In the context of 5G networks, WiGig can be used as a complementary technology to provide high-speed local area network (LAN) connections for mobile devices and IoT devices. The 60GHz frequency band has a limited range, but it can support high data rates over short distances, making it suitable for applications such as augmented and virtual reality (AR/VR), wireless HD video streaming, and cloud gaming [55]. In addition, WiGig can be used as a backhaul technology for small cells in 5G networks, enabling high-speed data transfers between small cells and the core network. This can help improve the performance and capacity of 5G networks, especially in densely populated urban areas where there is high demand for data services. Regarding IoT services, WiGig can enable high-speed local area connections between IoT devices, allowing them to share data quickly and efficiently. This can be especially useful for applications such as smart homes, where multiple IoT devices need to communicate with each other in real-time.

Thank you very much.

Sincerely,

Carolina Del-Valle-Soto

Universidad Panamericana. Facultad de Ingeniería. Álvaro del Portillo 49, Zapopan, Jalisco, 45010, México.

Phone: +52 (33) 13682200 | Ext. 4866

Reviewer 3 Report

Even if the topic is interesting and the first part of the paper seems promising, the second part is very naïve, contains a lot of errors (even related to very simple concepts), and misses a proper technical description of the features offered by the 5G technology to address the mentioned problems.

In the following, there are all the identified problems that involve a simple re-style of the text, small revisions of some sections, but also deep revisions of other sections:

·       Text within Figures 1, 3, 4, 6, and 7 is too small and has to be enlarged. It is almost impossible to read it!

·       Section 3.1: 5G actually works also in the Sub 6 GHz band. It has to be mentioned in this section.

·       Section 3.1: After mentioning Figure 4, it is better to list the 5G implementation requirements in a dotted list in order to better separate them instead of describing them in a single text paragraph without any new lines.

·       Section 3.3: the last sentence on page 10 ends with “and so on like the following”, but the first sentence on page 11 is not related to the same to it. Are one or multiple sentences missing? Check and correct this aspect.

·       Section 3.4: it is better to list the mentioned wireless communication technologies in a dotted list instead of multiple and too small sub-subsections.

·       Section 3.4.7 could become 3.5 since 5G applications are a different aspect than wireless communication technologies.

·       Section 3.6 (old 3.5 – 5G technologies): it is better to list the mentioned 5G technologies in a dotted list instead of multiple and too small sub-subsections.

·       Section 3.7 (old 3.6 – 5G problems): it is better to list the 5G problems in a dotted list in order to better separate them instead of describing them in a single text paragraph.

·       Use the same dotted list style for the Interferences and the Interferences in 5G.

·       Some of the interferences mentioned in Section 3.8.1 are synonyms even if are presented as different interferences. For example, inter-channel and inter-carrier are the same interference! While others are just correlated. Another example is inter-cell interference: “Inter-cell interference […] occurs when two users from neighboring cells attempt to use the same frequency band simultaneously” i.e., try to transmit using the same channel…

·       Heterogeneous networks (HetNet) are a kind of networks. It cannot be defined as a 5G standard! It is not a standard at all!

·       The sentence “The architectures include AdHocs, D2D, New radio, 5G NodeB, Hetnets, etc.” does not make sense at all! Device-to-Device (D2D) is a kind of communication, New Radio is the name of the 5G technology (like LTE is for 4G), 5G NodeB is the name of the 5G base station (link eNB is for 4G). None of these can be defined as architectures!

·       Generally speaking, there are a lot of repetitions. Most of the concepts are repeated multiple times, especially within Section 3. Besides, superfluous parts can be deleted. For example, the impact of interference is partially repeated also in Section 3.8.3. The history of mobile communication present in Section 3.8.3 does not add any additional value, since the paper just focuses on 5G and IoT.

·       Table 6 is not mentioned within the text.

·       Beamwidth is the physical amplitude of the beam. “This wider beam can supply applications with higher speeds”. This sentence does not make sense. They are two completely different and not related things! A wider beam allows covering a more wide area, but does not guarantee a higher data rate.

·       Real-time monitoring and management simply means monitoring and management in real-time….. it does not always imply a “remotely” component….

·       “QoS is a feature that 5G can provide”, “low latency is another feature that 5G can offer”, and “IoT devices can connect to the cloud more efficiently” are too naïve affirmations. How 5G can do this? Through which mechanisms and strategies? All section lacks a proper technical analysis of the analysed problem. All the mentioned concepts are just reports without proper technical and detailed discussions.

·       On page 27, “Table 5 provides an overview” should be Table 9

·       Section 4 is just a very superficial discussion that mainly summarises what the authors already wrote in the previous sections. No real/tangible and technological results are reported.

·       There are some typos throughout the text that have to be fixed, such as “devices can be allocated their own network resources” -> “devices can allocated their own network resources”, “allowed to creation of new protocols” -> “allowed to create new protocols”, “this section could cover” -> “this section covers”, “adjacent channel of our device uses overlaps” -> “adjacent channel of the one used by our device overlaps”, “internet devices of things” -> “internet of things devices” among many others.

Author Response

Dear

Editor

Sensors

We are submitting the paper:

“Utilization of 5G Technologies in IoT Applications: Current limitations by interference and network optimization difficulties - A Review”

Authored by: Mario Pons1, Estuardo Valenzuela1, Brandon Rodríguez1, Juan Arturo Nolazco-Flores2, and Carolina Del-Valle-Soto3,*

We would like to thank the reviewers and editors for their detailed analysis of the manuscript; the comments are very valuable to us. In the revised version of the paper, we have incorporated the all changes recommended by the reviewers.

Comments to all observations and suggestions including point-by-point responses are addressed in the following text.

Reviewer 3 comments

Comment 1: Even if the topic is interesting and the first part of the paper seems promising, the second part is very naïve, contains a lot of errors (even related to very simple concepts), and misses a proper technical description of the features offered by the 5G technology to address the mentioned problems.

In the following, there are all the identified problems that involve a simple re-style of the text, small revisions of some sections, but also deep revisions of other sections:

Text within Figures 1, 3, 4, 6, and 7 is too small and has to be enlarged. It is almost impossible to read it!

Response: Thanks to Reviewer. We have improved the font of the Figures you mention to make the paper more readable.

Comment 2: Section 3.1: 5G actually works also in the Sub 6 GHz band. It has to be mentioned in this section.

Response: The Reviewer is correct and we have included a substantial and well-supported paragraph in that Section with its respective reference.

5G also operates on two frequency bands: Sub 6 GHz and mmWave (millimeter wave). While mmWave offers faster data transfer rates, it has limited coverage and is easily obstructed. On the other hand, the Sub 6 GHz band, which includes frequencies below 6 GHz, offers wider coverage and can penetrate obstacles like buildings and walls [36]. This means that 5G can actually work in the Sub 6 GHz band, providing a more reliable and accessible network for users. This is particularly important for rural and suburban areas where building penetration is critical, and for indoor usage where higher frequency bands may not be able to reach. Therefore, despite the hype around mmWave, the Sub 6 GHz band remains a vital part of the 5G network, providing a strong foundation for the future of wireless technology.

Comment 3: Section 3.1: After mentioning Figure 4, it is better to list the 5G implementation requirements in a dotted list in order to better separate them instead of describing them in a single text paragraph without any new lines.

Response: The Reviewer's comment is very pertinent, and we have made the list that he/she proposes to give the manuscript more outstanding organization.

Based on the requirements above, the following are the 5G network deployment requirements:

  • Millimeter wave (mmWave) frequencies for higher bandwidth and faster data rates.
  • Massive MIMO (multiple input, multiple output) technology for increasing spatial streams and improving the wireless channel’s efficiency.
  • Beamforming to focus the radio signal towards a specific device, increasing signal strength and reducing interference.
  • Network slicing for creating dedicated virtual networks to meet the specific requirements of different applications, improving network performance and security.
  • Edge computing for real-time processing and analysis of data at the edge of the network, reducing latency and improving the performance of time-sensitive applications.
  • Low latency for real-time communication and response, aiming to achieve a latency of less than 1 millisecond.
  • High reliability for mission-critical applications, ensuring high availability and low downtime.
  • Full duplex communication for simultaneous transmission and reception of data, improving efficiency, capacity, and reducing latency

Comment 4: Section 3.3: the last sentence on page 10 ends with “and so on like the following”, but the first sentence on page 11 is not related to the same to it. Are one or multiple sentences missing? Check and correct this aspect.

Response: Thanks to Reviewer and we've added a paragraph that naturally connects both ideas.

The evolution of IoT technologies is having a significant impact on 5G networks. As more and more devices are connected to the internet, there is a growing need for faster, more reliable connectivity. This is where 5G networks come in, providing higher speeds, lower latency, and greater capacity than previous wireless technologies. The proliferation of IoT devices also creates new opportunities for 5G, as the technology is able to support the massive amounts of data generated by these devices. However, this also poses challenges for 5G networks, such as the need to handle large amounts of traffic from a variety of devices with different requirements. As a result, the evolution of IoT technologies is driving innovation in 5G networks, as providers look for ways to meet the demands of this growing ecosystem.

Comment 5: Section 3.4: it is better to list the mentioned wireless communication technologies in a dotted list instead of multiple and too small sub-subsections.

Response: What the Reviewer mentions is correct and we have organized that part of the paper into a list.

Comment 6: Section 3.4.7 could become 3.5 since 5G applications are a different aspect than wireless communication technologies.

Response: The Reviewer is right and they are a different look.

Comment 7: Section 3.6 (old 3.5 – 5G technologies): it is better to list the mentioned 5G technologies in a dotted list instead of multiple and too small sub-subsections.

Response: Many thanks to the Reviewer. All of his comments have made the manuscript more readable.

Comment 8: Section 3.7 (old 3.6 – 5G problems): it is better to list the 5G problems in a dotted list in order to better separate them instead of describing them in a single text paragraph.

Response: Thanks to the Reviewer for his pertinent comment.

The main problems mentioned and explained above are listed below:

  • Technical complications related to interference, including sensitivity to mild rain in urban areas.
  • Need for extensive and costly architectures to offer full coverage and optimal user experience, which may lead providers to focus mainly on urban areas, leaving rural areas unattended.
  • Ethical and social implications related to the inability to offer improved connections to people in poor conditions due to the high cost of 5G architectures and the lack of coverage in rural areas.
  • Security implications arising from the trend of connecting vehicles to the internet, which could result in remote kidnappings by hackers.

Comment 9: Use the same dotted list style for the Interferences and the Interferences in 5G.

Response: The Reviewer's comment has been corrected in the manuscript, which has given the paper much order.

Comment 10: Some of the interferences mentioned in Section 3.8.1 are synonyms even if are presented as different interferences. For example, inter-channel and inter-carrier are the same interference! While others are just correlated. Another example is inter-cell interference: “Inter-cell interference […] occurs when two users from neighboring cells attempt to use the same frequency band simultaneously” i.e., try to transmit using the same channel…

Response: The Reviewer is right, and we have omitted Inter-carrier interference not to confuse the reader. We have omitted it from the text and the table. We have left the other classification that was conceptually more general.

Comment 11: Heterogeneous networks (HetNet) are a kind of networks. It cannot be defined as a 5G standard! It is not a standard at all!

Response: The Reviewer is correct and we have corrected the error in the text.

Comment 12: The sentence “The architectures include AdHocs, D2D, New radio, 5G NodeB, Hetnets, etc.” does not make sense at all! Device-to-Device (D2D) is a kind of communication, New Radio is the name of the 5G technology (like LTE is for 4G), 5G NodeB is the name of the 5G base station (link eNB is for 4G). None of these can be defined as architectures!

Response: Thanks to Reviewer. We have completely omitted that phrase.

Comment 13: Generally speaking, there are a lot of repetitions. Most of the concepts are repeated multiple times, especially within Section 3. Besides, superfluous parts can be deleted. For example, the impact of interference is partially repeated also in Section 3.8.3. The history of mobile communication present in Section 3.8.3 does not add any additional value, since the paper just focuses on 5G and IoT.

Response: Many thanks to the Reviewer for pointing out these aspects that give order and coherence to the manuscript. We omitted so much repetition of "impact of..." and history and evolution that had nothing to do with the technical part.

Comment 14: Table 6 is not mentioned within the text.

Response: We have added an explanatory paragraph about the Table so that it is mentioned in the text.

Comment 15: Beamwidth is the physical amplitude of the beam. “This wider beam can supply applications with higher speeds”. This sentence does not make sense. They are two completely different and not related things! A wider beam allows covering a more wide area, but does not guarantee a higher data rate.

Response: We thank the Reviewer. We have corrected the error and clarified that we were not referring to beamwidth but to bandwidth. We have corrected it in the four appearances it had in the paper.

Bandwidth refers to the amount of data that can be transmitted over a network or a communication channel in a given amount of time. In the context of 5G, higher bandwidth means that more data can be transmitted over the airwaves in a shorter period of time, which can result in faster and more reliable 5G transmissions.

Comment 16: Real-time monitoring and management simply means monitoring and management in real-time….. it does not always imply a “remotely” component….

Response: We have corrected and omitted the word "remotely" to avoid inaccuracies, as well mentioned by the Reviewer.

Comment 17: “QoS is a feature that 5G can provide”, “low latency is another feature that 5G can offer”, and “IoT devices can connect to the cloud more efficiently” are too naïve affirmations. How 5G can do this? Through which mechanisms and strategies? All section lacks a proper technical analysis of the analysed problem. All the mentioned concepts are just reports without proper technical and detailed discussions.

Response: Thanks to the Reviewer.

Regarding QoS, we have tried to clarify with the following paragraph:

In the context of IoT connections, QoS refers to the ability of the network to provide reliable and high-performance connectivity to IoT devices. 5G is a promising technology for IoT connections as it can provide several features that can improve the overall QoS and performance of IoT applications. 5G is designed to provide a significantly improved QoS compared to previous generations of mobile networks. This is achieved through a combination of advanced technologies and features that are built into the 5G standard. One of the key features of 5G that can contribute to a better QoS is the use of advanced radio access technologies. These technologies, such as massive MIMO and beamforming, allow for more efficient use of the radio spectrum and better signal quality. This means that 5G can provide better connectivity and signal strength, which can improve the reliability and availability of the network. Another important feature of 5G that can contribute to a better QoS is the use of network slicing. Network slicing allows the network to be divided into multiple virtual networks, each tailored to specific use cases and requirements. This means that different applications and services can be given different levels of priority, bandwidth, and latency requirements, depending on their needs. This can improve the QoS for each individual application and user. In addition, 5G can provide lower latency, higher throughput, and better support for massive IoT deployments, all of which can contribute to a better QoS. Lower latency means that there is less delay between the transmission of data and its reception, which can enable real-time communication and faster response times. Higher throughput means that more data can be transmitted over the network in a given time, which can improve the performance of applications that require high-bandwidth data transfer. Finally, better support for massive IoT deployments means that the network can handle a large number of connected devices and data traffic, without compromising the QoS for individual devices or applications.

Regarding latency and IoT devices, we have tried to improve the ideas with the following paragraphs:

Low latency is particularly important for applications that require immediate decision-making, such as autonomous vehicles or remote surgery. With 5G, the latency can be reduced to as low as one millisecond, which can enable real-time communication and faster response times. Furthermore, 5G can provide higher throughput, which refers to the amount of data that can be transmitted over a network in a given time. With higher throughput, IoT devices can transmit and receive larger amounts of data, which can improve the performance of IoT applications that require high-bandwidth data transfer, such as video streaming or real-time sensor data. Cloud computing is another feature that 5G can offer for IoT connections. With 5G, IoT devices can connect to the cloud more efficiently, and each device can use all the resources of the cloud, such as storage and processing power. This feature can significantly improve the performance and scalability of IoT applications, especially those that require large amounts of data processing and storage.

Comment 18: On page 27, “Table 5 provides an overview” should be Table 9

Response: We have corrected this naming error.

Comment 19: Section 4 is just a very superficial discussion that mainly summarises what the authors already wrote in the previous sections. No real/tangible and technological results are reported.

Response: Many thanks to the Reviewer. We have added the following tables and figures to complement the discussion, delving into comparative analyses showing an analytical contribution at the current literature review level.

In recent years, the deployment of 5G networks has gained significant attention due to its potential to revolutionize the communication industry. One of the areas where 5G networks are expected to have a substantial impact is the IoT services. This literature review article aims to provide an in-depth analysis of the impact of 5G networks on IoT services, specifically examining the issue of interference in this type of network and its related technologies.

We have also added more explanatory tables, which demonstrate the comparative analysis requested by the Reviewer.

Comment 20: There are some typos throughout the text that have to be fixed, such as “devices can be allocated their own network resources” -> “devices can allocated their own network resources”, “allowed to creation of new protocols” -> “allowed to create new protocols”, “this section could cover” -> “this section covers”, “adjacent channel of our device uses overlaps” -> “adjacent channel of the one used by our device overlaps”, “internet devices of things” -> “internet of things devices” among many others.

Response: Many thanks to the Reviewer. We have reviewed the manuscript, however, if necessary we will contract the English editing service of the Journal.

Thank you very much.

Sincerely,

Carolina Del-Valle-Soto

Universidad Panamericana. Facultad de Ingeniería. Álvaro del Portillo 49, Zapopan, Jalisco, 45010, México.

Phone: +52 (33) 13682200 | Ext. 4866

Round 2

Reviewer 3 Report

The authors carefully read my comments and revised the manuscript accordingly. The quality is much improved and the paper is now ready to be published.